# Altered predictive control during memory suppression in PTSD

Giovanni Leone [1], Charlotte Postel[1], Alison Mary [1,2], Florence Fraisse [1], Thomas Vallée [1], Fausto Viader [1], Vincent de La Sayette[1], Denis Peschanski [3], Jaques Dayan[1,4], Francis Eustache [1] & Pierre Gagnepain [1✉]

Aberrant predictions of future threat lead to maladaptive avoidance in individuals with post-traumatic stress disorder (PTSD). How this disruption in prediction influences the control of memory states orchestrated by the dorsolateral prefrontal cortex is unknown. We combined computational modeling and brain connectivity analyses to reveal how individuals exposed and nonexposed to the 2015 Paris terrorist attacks formed and controlled beliefs about future intrusive re-experiencing implemented in the laboratory during a memory suppression task. Exposed individuals with PTSD used beliefs excessively to control hippocampal activity during the task. When this predictive control failed, the prediction-error associated with unwanted intrusions was poorly downregulated by reactive mechanisms. This imbalance was linked to higher severity of avoidance symptoms, but not to general disturbances such as anxiety or negative affect. Conversely, trauma-exposed participants without PTSD and nonexposed individuals were able to optimally balance predictive and reactive control during the memory suppression task. These findings highlight a potential pathological mechanism occurring in individuals with PTSD rooted in the relationship between the brain's predictive and control mechanisms.

[1] Normandie Univ, UNICAEN, PSL Research University, EPHE, INSERM, U1077, CHU de Caen, GIP Cyceron, Neuropsychologie et Imagerie de la Mémoire Humaine, 14000 Caen, France. [2] Neuropsychology and Functional Imaging Research Group (UR2NF), Centre for Research in Cognition and Neurosciences (CRCN), UNI – ULB Neuroscience Institute, Université libre de Bruxelles (ULB), Brussels, Belgium. [3] Université Paris I Panthéon Sorbonne, HESAM Université, EHESS, CNRS, UMR8209 Paris, France. [4] Pôle Hospitalo-Universitaire de Psychiatrie de l'Enfant et de l'Adolescent, Centre Hospitalier Guillaume Régnier, Université Rennes 1, 35700 Rennes, France. ✉email: pierre.gagnepain@inserm.fr

ndividuals with post-traumatic stress disorder (PTSD) avoid traumatic reminders in order to anticipate threat[1,2] and reduce distress. Their perception of the future may have changed in the aftermath of the traumatic experience[3,4]. Bayesian models of the brain[5] provide a solution to understand this impairment in predictive processing[6]. More specifically, aberrant associations may arise between safe environmental cues and threatening outcomes[5,7], thereby compromising their ability to accurately predict aversive events[8]. This disruption in prediction exacerbates the avoidance of trauma reminders[1], which may prevent the extinction or updating of the traumatic engram. The impact of this disruption in prediction on the control of the re-experiencing of unintentional flashbacks or intrusive memories (i.e., cardinal symptom of PTSD[9]), however, is unknown.

In a recent study, we suggested that the persistence of intrusive memories in individuals with PTSD may be rooted in a generalized dysfunction of the inhibitory control system that normally regulates unwanted memories[10]. In this study, 102 participants who had been exposed to the November 2015 Paris terrorist attacks, as well as 73 nonexposed individuals, learned a series of neutral words paired with images of objects, and were later instructed to suppress the unwanted re-experiencing of intrusive memory images involuntarily triggered by the word reminder cue. During this suppression phase, we recorded brain activity using functional magnetic resonance imaging (fMRI), and participants were asked to report the presence or absence of intrusions at each trial. Exposed participants were divided into two subgroups: individuals with PTSD symptoms, and resilient individuals who did not develop PTSD. Resilient individuals exhibited a decrease in functional coupling between control and memory brain networks during the experiencing of intrusive memories, compared with both nonintrusive and resting-state conditions. This pattern is consistent with an increase in inhibitory (i.e. negative) coupling during suppression of intrusive memories. Dynamic causal modeling (DCM) analyses confirmed that this decrease in coupling reflected top-down mechanisms orchestrated by the right dorsolateral prefrontal cortex (DLPFC)[11]. In memory regions involved in the persistence of the trauma, such as the hippocampus and precuneus (PC)[12], this controlled down-regulation of intrusive memories was severely compromised in individuals with PTSD, whose brain dynamics did not differ between the intrusive and nonintrusive conditions.

These findings highlight a fundamental role of memory control mechanisms in the development of PTSD in response to trauma, but tell us nothing about the origin of their disruption and the potential contribution of hidden computations underlying predictions of intrusive memories. Cognition, motor responses and memories can be controlled by an early proactive mechanism that biases attention according to goals, and additionally corrected during a late reactive process[13,14]. Interestingly, these processes are captured well by Bayesian models that incorporate the dynamic adjustment of predictions based on previous experiences and the use of prediction error (PE) to modulate the future need for control and its correction[15]. The prediction-based dynamic adjustment of the forthcoming need for control reflects a form of *predictive control* that critically depends on the DLPFC[16]. We hypothesized that the inhibitory control of memories also relies on predictive inferences, and that the interaction between predictive and control processes is central to understanding the pathogenesis of PTSD.

We can assume that estimated probabilities of intrusive re-experiencing based on prior encounters (i.e., beliefs) are aberrantly prioritized in individuals with PTSD[17], such that control resources are allocated to a form of predictive avoidance that overrides online memory signals. For instance, individuals with PTSD may not only avoid situations for which they anticipate

flashbacks, such as certain places or times of the day, but may also use this expectation to proactively alter conscious thoughts[18]. Alternatively, the reduced inhibitory control in individuals with PTSD may be limited to reactive processes targeting the online emergence of intrusive memories, given their hypersensitivity to PE[5] which may reduce the control resources available and inhibitory coupling. In the context of a memory suppression task, like other situations requiring flexible cognitive control, prior exposure to successive reminders influences the belief that an undesired memory will emerge into consciousness while processing the upcoming cue[15]. Critically, exaggerated predictive control, reduced reactive control, or a combination of the two may explain our previous observation that the brain connectivity markers of memory suppression are disrupted in individuals with PTSD (see Fig. 1b)[10].

In the current study, we tracked these hidden computations during the think/no-think (TNT) memory suppression task using meta-Bayesian modeling[19] and analyzed their impact on the underlying connectivity markers of memory control. We applied this analysis to the same subgroups with (PTSD+; $n = 55$) or without (PTSD−; $n = 47$) PTSD following exposure to the terrorist attacks in Paris on 13 November 2015, and the same nonexposed participants ($n = 73$)[10] (see the "Methods" section). We submitted trial-by-trial computations of beliefs about upcoming intrusions and resulting PE to a DCM analysis to explore their influence on the effective connectivity between the inhibitory control system and memory target regions. We focused this analysis on the right anterior and posterior middle frontal gyrus (MFG)[10,20], two core nodes of the inhibitory control system, and tested their relative contribution to belief-driven and PE-driven control. We tested the influence of these two distinct control hubs on two memory regions that are central to the establishment of traumatic memory: the hippocampus, distinguishing between its rostral and caudal parts[10] and the PC.

## Results

**Computational modeling**. To track beliefs about upcoming intrusive memories, we applied three distinct models of increasing complexity (Fig. 1c): (1) the Rescorla–Wagner (RW)[21], which postulates that trial-by-trial PE updates beliefs at a fixed learning rate; (2) the Kalman filter (KF)[22], in which the updating of beliefs relies on a dynamic (i.e., not fixed) learning rate, shaped by additional trial-by-trial uncertainty weighting of PE, assuming that such uncertainty is constant and the learned environment not volatile; and (3) the two-level hierarchical Gaussian filter (HGF)[23] which, like the KF, assumes that the learning underlying belief updating is a dynamic process based on uncertainty, but further assumes that the environment is volatile, and which involves the hierarchical embedding of beliefs. Note, however, that the two-level HGF model can also be interpreted as a Kalman filter operating at the (logit-transformed) contingency level as opposed to simply the outcome level like our current implementation of the KF model.

We built three distinct source models to map intrusion beliefs onto outcome probabilities. Each of these models assumed different sources of beliefs, in order to establish their accuracy in predicting the outcome. The *state* source model assumed that participants formed beliefs based solely on the trial history. The *item* source model assumed that beliefs were based solely on the history of each specific word–object pair (see Fig. 2a), disregarding overall trial-by-trial history. The *combined* source model assumed that the combination of state and item (precision-weighted) beliefs improves prediction accuracy (see Eq. (14) and Fig. 2a). These three source models mapped beliefs onto binary ratings through a beta function, with a free parameter estimating

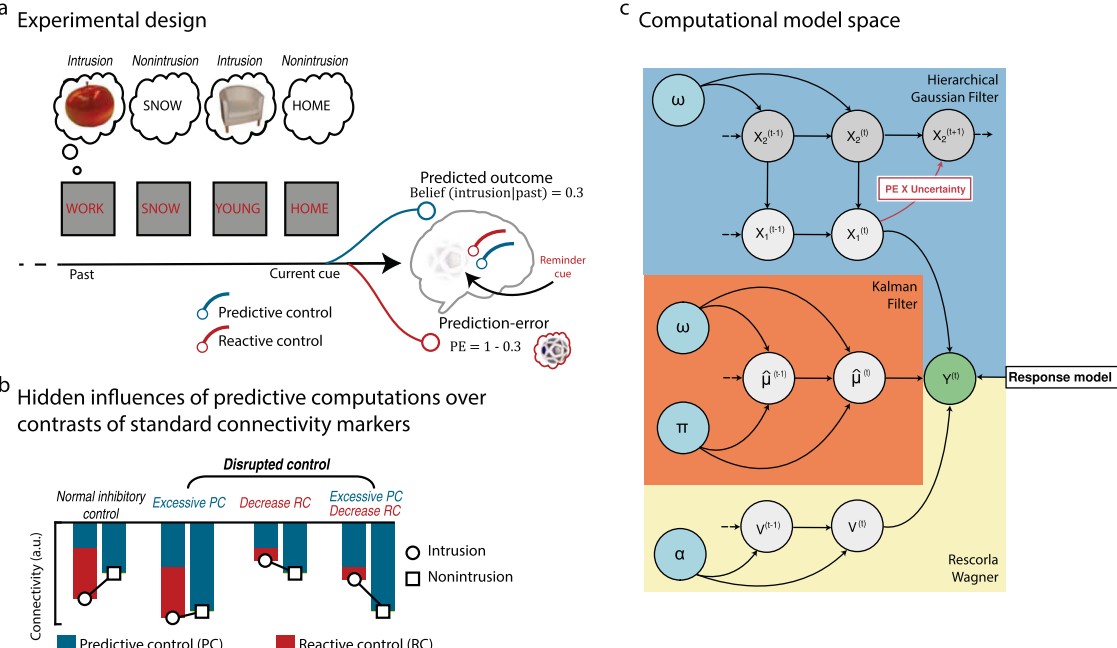

**Fig. 1 Design and computational models. a** After learning word–object pairs, participants performed a memory suppression task in which they were asked to prevent the memory of the images associated with the cue words from entering awareness. They then rated the presence or absence of intrusive memories during suppression attempts. The estimation that an upcoming cue will trigger an intrusive memory (i.e., belief) can be inferred from previous encounters, providing an adaptive advantage in the form of the deployment of optimum memory control and proactive prevention of memory retrieval (i.e., *predictive control*). Reactive control is engaged when intrusive memories unexpectedly cross the proactive gate, resulting in a prediction error (PE) that triggers additional inhibition and updating of future expectations. It should be noted that recall cues (i.e., think items) are not displayed here (see the "Methods" section). The apple and the chair items are selected from the Bank Of Standardized Stimuli (BOSS) and published under CC BY SA license (https://creativecommons.org/licenses/by-sa/3.0/)[55]. **b** Toy example. Standard contrast analyses of intrusive and nonintrusive cues cannot identify the contribution of these critical computational quantities on the disruption of the connectivity markers of inhibitory control. **c** Computational model space. Binary intrusion ratings across the suppression task were fed into computational models to track belief formation across the suppression task. In the two-level hierarchical Gaussian filter (HGF; pale blue panel), beliefs are hierarchical and dynamically weighted by uncertainty. The perceptual parameter $\omega$ regulates the speed of belief adjustment throughout the task. The Kalman filter (KF; pale orange) also includes dynamic belief updating, which is regulated by two free perceptual parameters, $\pi$ and $\omega$, encoding belief reliability and uncertainty, but it does not assume hierarchical beliefs. The Rescorla–Wagner model (RW; pale yellow) is a simpler non-hierarchical model with a fixed, participant-specific learning rate $\alpha$. The response model describes the log-probability of the behavioral outcomes (i.e., intrusion or nonintrusion rating) given beliefs through a beta density function. These trial-wise log-probabilities are used to compute model accuracy.

the accuracy of this mapping (see the "Methods" section). Model accuracy was computed using the negative log-likelihood of the choice probability for each of the nine computational models (HFG-state, HFG-item, HFG-combined, RW-state, RW-item, RW-combined, KF-state, KF-item, KF-combined).

*Model validation.* We performed different simulations to determine whether our model produced valid and reliable outputs. Intrusion ratings decreased across blocks of trials in the TNT task[10]. We first performed model falsification[24] to evaluate whether our computational models could generate this expected pattern of behavioral responses across a wide range of simulated model parameters. This analysis is reported in detail in the "Methods" section, but briefly, consisted in simulated synthetic beliefs from 200 virtual participants using the above-mentioned models, and repeated the virtual experiment 100 times using perceptual parameter randomly drawn from a Gaussian priors distribution tailored to match our own data (to sample plausible parameters), resulting in 20,000 simulations for each of the nine computational models. Then, synthetic beliefs were mapped into binary ratings which were averaged across repeated sampling and summarized as intrusion proportion across the four artificial TNT sessions (see Fig. 2a). Second, we tested for each model whether we could recover the simulated trajectories of beliefs,

and whether these trajectories were distinguishable among competing source models. We fit synthetic binary data generated with the same, as well as competing, source models (i.e. state, item, and combined), and compared the resulting trajectories to simulated ones using correlation. Results revealed we could confidently recover the true generated trajectories among competitor source models for HGF, but not for RW or KF models (see Fig. 2b). Third, we use the same logic to verify the reliability of the model selection criterion for identifying the true generative model within a set of competitive source models, and ensure that this selection is not biased in favor of one particular model[24,25]. This procedure, known as model recovery, consists in simulating data with one specific model and then comparing the predictive performances (i.e. model accuracy) of a set of different models. This analysis confirmed that the comparison between these three source models was not biased for HGF (Fig. 2c). However, the probability of recovering the true model was confounded with competing source models for RW and KF (Fig. 2c). Fourth, we performed parameter recovery analyses[25], to ensure the reliability and meaningfulness of estimated model perceptual parameters. Results of these analyses, reported in detail in the "Methods" section (see also Fig. 2d), indicated that parameter recovery was modest for the HGF model and poor for RW or KF.

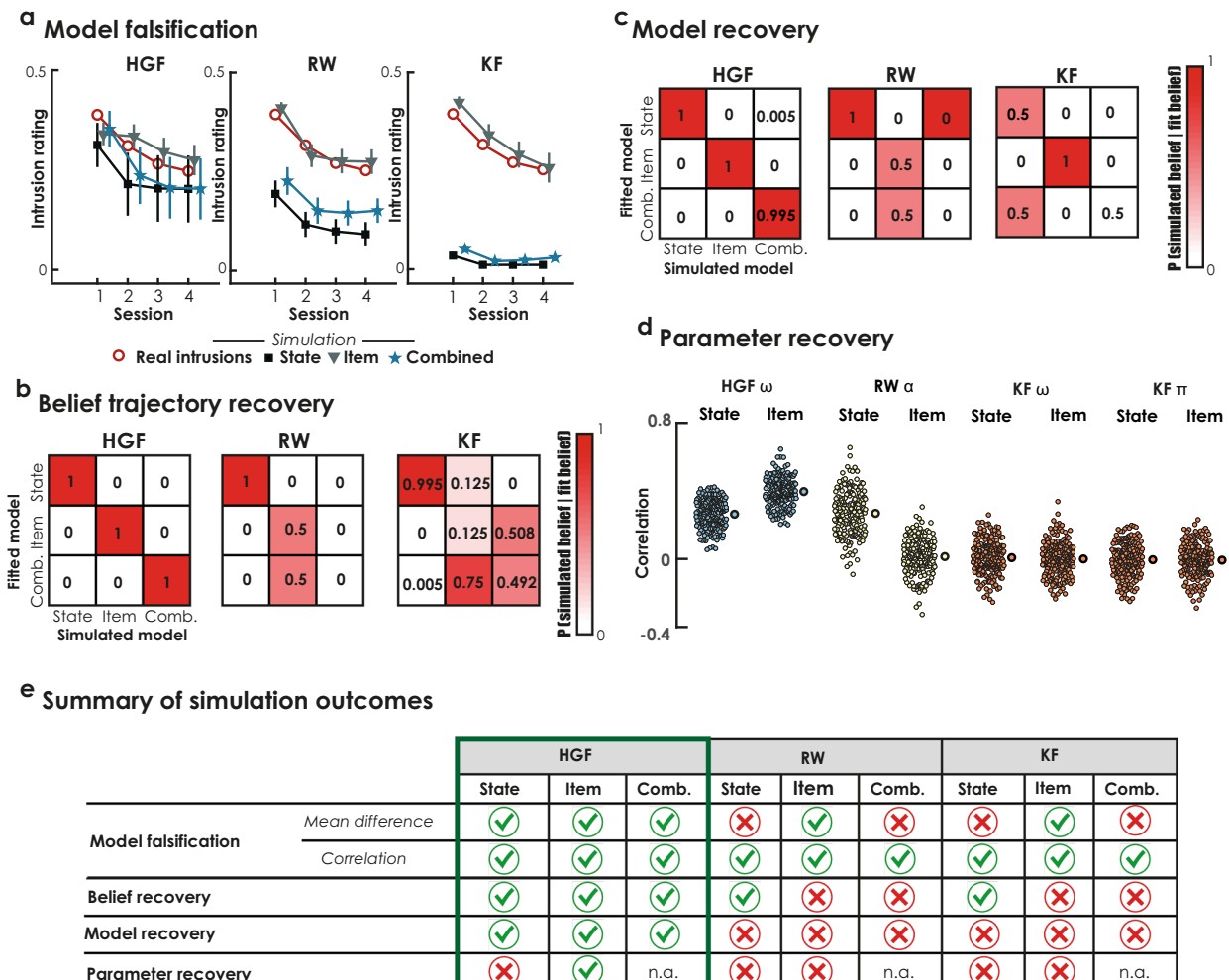

**Fig. 2 Validation of computational models. a** Model falsification. In order to test the models' generative performance (i.e., the model ability to generate plausible data), we generated synthetic intrusion data for each model, simulating 200 virtual participants for which we repeated the simulations 100 times (so 20,000 simulations in total). We reported the session-wise mean trajectories of real intrusions rating (empty red circles) and simulated intrusions data, under HGF, RW and KF models, for both state (black squares), item (gray triangles), and combined (blue stars) sources model versions. Error bar represents 95% confidence intervals of the virtual participants' distribution. **b** Belief recovery. Inversion matrix reflecting the confidence that the beliefs fitted by a given model was the model that most likely has generated those beliefs. **c** Model recovery. Inversion matrix reflecting the confidence that the best fitting model has generated the data. **d** Parameter recovery. Correlation between fitted and simulated model parameter for each virtual participant and each model. The large dot at the right of each distribution represents the mean correlation across virtual participants. **e** Summary of simulation outcomes.

In summary, in the current experimental setting, only the HGF model produced valid trajectories of intrusion beliefs, which accurately simulate the behavioral pattern, and reliably and truthfully distinguished beliefs formed on the basis of the trial or item history, or a combination of both memory sources. We therefore used the HGF to track trial-by-trial variations in beliefs about the potential re-experiencing of upcoming intrusive memories attached to a cue word and the resulting PE, and investigate the influence of these estimates on brain control mechanisms using connectivity analysis. The perceptual parameter ($\omega$) for this model is a participant-specific constant indicating the speed at which these beliefs are changing. We then tested whether our model was sufficiently powered to detect changes in this parameter. To test this, we simulated and recovered parameters for two distinct synthetic groups using an effect size in a range of our data (i.e. the average difference in perceptual parameter between groups) and then performed statistical tests to detect group differences in this simulated data set. The statistical power to detect group difference on the model perceptual parameter (corresponding to the frequency of

significant test in this simulated data sets) was 90% for HGF-item, 10% for HGF-state. This suggests that this perceptual parameter can be confidently recovered from intrusion beliefs and compared between groups when it is derived from the item structure, but not from the task state (note, however, that the outcomes of the following analysis of connectivity are independent, and not related to this perceptual parameter; see Supplementary Fig. 1). Regarding the $\omega$ parameter computed for item beliefs, the PTSD+ group expressed significantly slower beliefs updating than the nonexposed group, $t(122) = -2.10$, $p = 0.037$, bootstrapped 95% CI [−0.59, 0.06], and a trend compared with the PTSD− group, $t(99) = 1.82$, $p = 0.072$, bootstrapped 95% CI = [−0.58, −0.04]; although this effect was significant when the bootstrapping of the mean is considered). No differences in item belief updating were found between the nonexposed and PTSD− groups, $t(113) = 0.15$, $p = 0.880$, bootstrapped 95% CI [−0.22, 0.27]. Compared to nonexposed controls and participants without PTSD, individual with PTSD were less prone to shift their beliefs about a particular item after they failed to control it and suppress the associated intrusion.

*Source of intrusion beliefs.* To determine the memory source of intrusion beliefs (i.e. state, item, or combined), we performed Bayesian model selection (BMS) and compared the accuracy of the three source models at the population level. This analysis revealed that the combined model (*protected* exceedance *probability*, PXP = 0.999) outperformed the other two source models (Bayesian omnibus risk; see the "Methods" section, BOR = 0). The probability that the same model would optimally explain data in all three groups $P(H_{F=}|y)$ was 0.996 (see the "Methods" section).

Taken together, these findings suggest that, in all three groups, beliefs about the experiencing of memory intrusions across suppression attempts (1) spread according to a two-level hierarchy that took volatility of beliefs uncertainty into account, (2) were driven by a flexible and dynamic learning process updated by PE, and (3) originated from the merging of recent *meta-memories* about their control performance that derived from both trial history and item-specific memories, as observed in other forms of cognitive control[26].

**Computational dynamic causal modeling.** For each cue word, our combined HGF2 computational model provided an estimate of the participant's hidden belief that the cue would trigger an intrusive memory, as well as an estimate of the discrepancy (i.e., PE) between the expected and experienced outcome (see Figs. 1c and 2a). We then investigated the influence of these estimates on brain control mechanisms, using DCM. We distinguished predictive mechanisms engaged to suppress intrusion beliefs from reactive mechanisms related to the additional demand of controlling the error induced by intrusive memories. For instance, if a cue was associated with an intrusion belief ($\hat{\mu}_1^{(t)}$) of 0.3, then the presence of an intrusion ($y^{(t)} = 1$) would require additional PE control of 0.7 ($PE^{(t)} = y^{(t)} - \hat{\mu}_1^{(t)}$), see Fig. 1a. These quantities were used as parametric modulators of the inputs (i.e., stick function) modulating the top-down coupling between control and memory systems. It should be noted that we focused this analysis on positive PE (PE+) to specifically isolate reactive control associated with suppression, and discarded negative PE associated with the absence of control demands during non-intrusive cues. However, parametric modulation of belief was performed for all cues.

We built 42 DCM models, which could be divided into three families expressing different hypotheses on the involvement of these computations. The first family, corresponding to our main hypothesis, assumed that these computations influenced top-down control. A second family tested the influence of these computations on bottom-up connections. A third family, in which the modulatory stick function of suppression trials was not parametrically modulated, tested the absence of influence of these computations on top-down control (i.e., no-computation models). Each of these families included reciprocal hypotheses about the role of the anterior MFG (aMFG) and posterior MFG (pMFG) in predictive and reactive control (see Fig. 3a). Half the models were assigned to the predictive or reactive influence of the pMFG and aMFG, and the other half to the opposite relationship. These six subfamilies therefore each contained seven models describing the possible combinations of modulation pathways between the MFG and the target regions (see Fig. 3b). Target regions included the rostral hippocampus (rHIP) and caudal hippocampus (cHIP), as well as the ventral portion of the PC (see the "Methods" section for the definition of volumes of interest and timecourse extraction). In addition to these 42 models testing our main hypotheses, we included a null model family hypothesizing an absence of controlled modulation (see Fig. 3a).

*Combined influence of anterior and posterior MFG during control.* First, we investigated whether beliefs and PE+ effectively modulated the causal influence of MFG on memory regions across all groups. In other words, we wanted to know whether predictive and reactive control mechanisms could explain the top-down coupling between these regions during motivated forgetting. Accordingly, the 14 models assuming a top-down modulation of control by belief and PE+ (i.e., first family), were compared with the models belonging to the bottom-up, no-computation and null families. We found overwhelming evidence (PXP = 0.886) that these computational quantities influenced top-down modulation, whereas the bottom-up (PXP = 0), no-computation (PXP = 0.113), and null (PXP = 0) hypotheses (fBOR = 0) were not validated. The probability that the model frequency in favor of top-down computational models was the same for all three groups in our sample was equal to $P(H_{F=}|y) = 0.968$.

After showing the top-down controlled modulation of belief and PE+, we asked whether the aMFG and pMFG were differentially involved in these two distinct mechanisms. BMS revealed no clear evidence in favor of one family over the other (PXP = 0.343 and PXP = 0.657, fBOR = 0.677). Further between-group comparisons revealed that the probability that there were no underlying differences in model architecture was equal to $P(H_{F=}|y) = 0.828$ when PTSD+ and PTSD− groups were compared, and $P(H_{F=}|y) = 0.796$ when PTSD+ and nonexposed groups were compared.

*Excessive belief suppression and alteration of reactive control in PTSD.* To compare reactive and predictive control mechanisms between groups, we performed Bayesian model averaging (BMA) of the 14 models included in the computational top-down family for each group separately. This was possible because the DCM architecture that best explained our data was the same across all three groups. However, given that no differences were observed within the combined family, we summed the coupling parameters from aMFG and pMFG to reflect the coordinate action of the core control network. BMA provides both individual- and group-specific posterior distribution of coupling parameters, weighted for posterior evidence across all models in a family (see the "Methods" section).

Our main hypothesis was that individuals with PTSD prioritize belief of intrusive memories over online re-experiencing (PE+), to proactively suppress memory processing (i.e., imbalance hypothesis). A marker of suppression has been associated with more pronounced top-down negative coupling[11,27]. We therefore expected the imbalance in individuals with PTSD to be associated with more negative coupling during predictive versus reactive control. We computed the interaction between control (i.e., predictive vs. reactive) and group (PTSD+ vs. PTSD− or nonexposed). We found disproportionate negative coupling with the rHIP during predictive versus reactive control in the PTSD+ group, compared with the nonexposed group, $t(125) = -2.81$, $p_{\text{false discovery rate, FDR}} = 0.007$, posterior probability (Pp) = 0.999, 95% CI [−0.99, −0.17], and PTSD-, $t(99) = -2.17$, $p_{\text{FDR}} = 0.009$, Pp = 0.999, 95% CI [−1.01 −0.01]. We found a similar Control *Group interaction for the cHIP, when we compared PTSD+ with PTSD−, $t(99) = -3.23$, $p_{\text{FDR}} = 0.006$, Pp = 1, 95% CI [−1.20, −0.29], and a trend toward significance when we compared PTSD+ with the nonexposed group, $t(125) = -1.62$, $p_{\text{FDR}} = 0.071$, Pp = 0.995, 95% CI [−0.73, 0.07]. The same pattern emerged when we combined the two parts of the hippocampus (i.e., wHIP), with PTSD+ showing a greater imbalance between predictive and reactive control than the nonexposed, $t(125) = -2.49$, $p_{\text{FDR}} = 0.014$, Pp = 0.992, 95% CI [−0.81, −0.09], and PTSD−, $t(99) = -2.91$, $p_{\text{FDR}} = 0.007$,

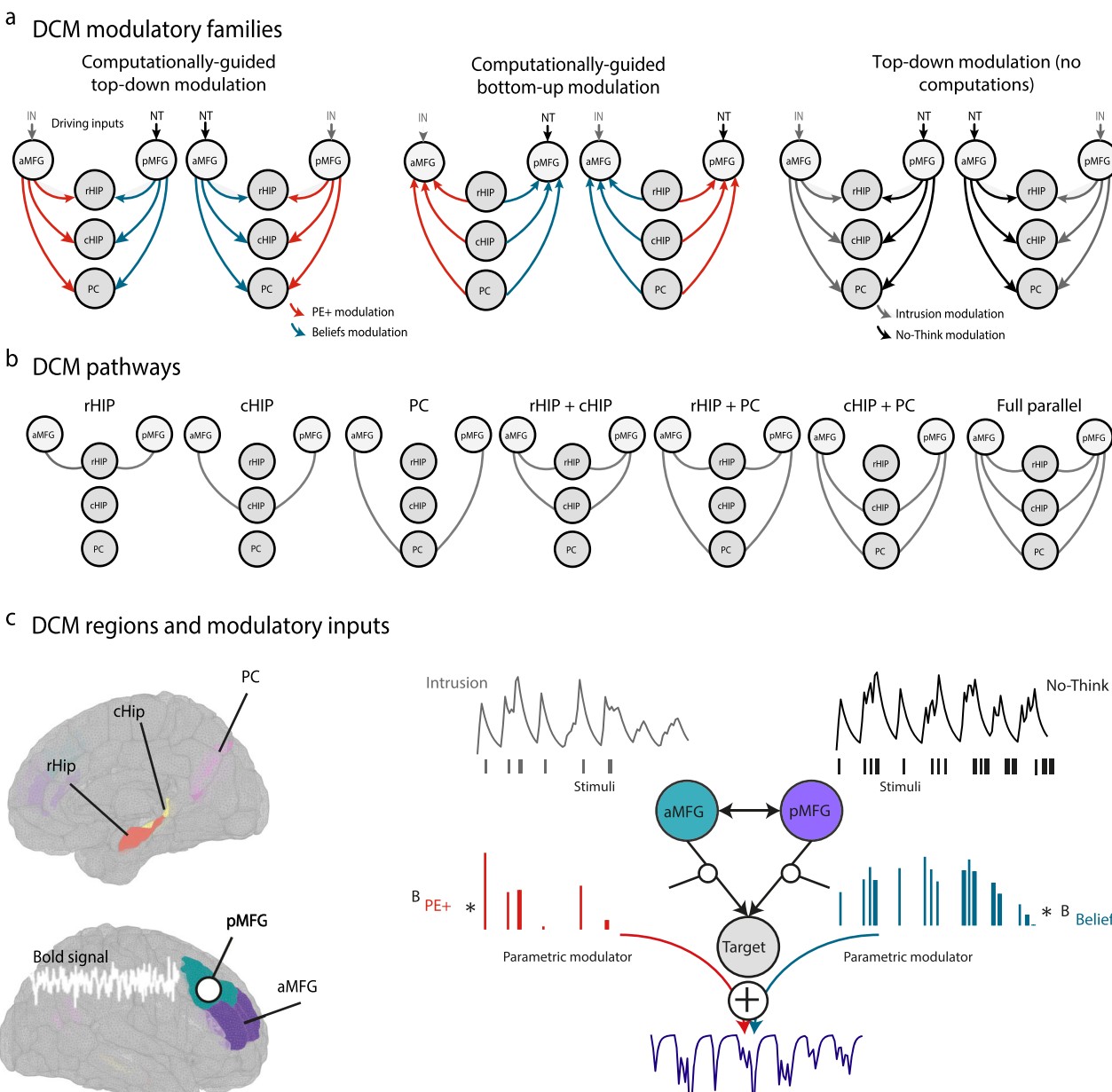

**Fig. 3 DCM models. a** DCM model families expressing different hypotheses on the involvement of intrusion beliefs and PE+ computations in the modulation of the coupling between control regions (anterior and posterior middle frontal gyrus, MFG) and memory target regions, including the rostral hippocampus (rHIP), caudal hippocampus (cHIP), and precuneus (PC). It should be noted that null models were also estimated, but are not shown here. **b** Pathways capturing the seven possible connections between control and target regions. **c** Left panel shows the regions of interest used for DCM analysis. Right panel provides an illustration of the modularity inputs influencing the connectivity between brain regions.

Pp = 0.998, 95% CI [−1.06, −0.19], groups. No differences were found in the PC when the PTSD+ group was compared with the PTSD−, $t(99) = -0.05$, $p_{FDR} = 0.477$, Pp = 0.540, 95% CI [−0.48, 0.50], and nonexposed, $t(125) = 0.13$, $p_{FDR} = 0.477$, Pp = 0.586, 95% CI [−0.42, 0.37], groups.

To further characterize these interactions, we explored the main effect of control and the simple effects of coupling parameters, running $t$ tests for each group and each target region. Statistical details of these analyses are reported in Table 1, as well as in Fig. 4. In summary, we observed significant negative coupling during reactive control of the hippocampus in both the nonexposed and PTSD− groups, but not in the PTSD+ group. By contrast, predictive control over the hippocampus was observed in all three groups. When we compared predictive and reactive

control within each group, we found significant higher inhibitory control of beliefs compared with PE, but only for the PTSD+ group in the rHIP, cHIP and wHIP. No differences were found in the other two groups (see Fig. 4 and Table 1). The PC was controlled proactively, but not reactively, in all three groups.

*Excessive predictive control is related to re-experiencing and avoidance dimensions of PTSD but not transdiagnostic symptoms.* We then examined whether the excessive of predictive control observed in individuals with PTSD could be specifically related to re-experiencing and avoidance symptoms, the two dimensions of PTSD presumably associated with such disruption, rather than to the general alteration of mental health. While intrusion and avoidance are two cardinal features of PTSD related to the

**Table 1 Within-group BMA coupling parameter statistics.**

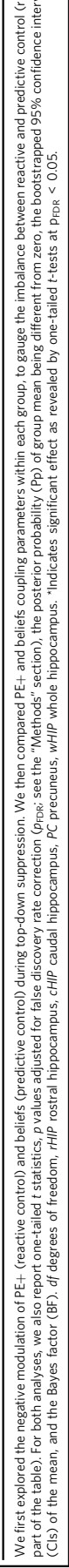

| | | df | Reactive control (PE+) | | | | | Predictive control (beliefs) | | | | | Predictive – reactive control | | | | |
|---|---|---|---|---|---|---|---|---|---|---|---|---|---|---|---|---|---|
| | | | t | pFDR | Pp | 95% CI | BF | t | pFDR | Pp | 95% CI | BF | t | pFDR | Pp | 95% CI | BF |
| Nonexposed | rHIP | 71 | −2.88 | 0.012* | 0.999 | [−0.37 −0.07] | 9 | −5.04 | <0.001* | 1 | [−0.46 −0.21] | >1000 | −0.98 | 0.178 | 0.884 | [−0.31 0.10] | 2.5 |
| | cHIP | 71 | −1.84 | 0.085 | 0.968 | [−0.22 0.01] | 108 | −4.02 | <0.001* | 1 | [−0.28 −0.16] | >1000 | −1.77 | 0.068 | 0.989 | [−0.41 0.01] | 7.4 |
| | PC | 71 | −0.05 | 0.478 | 0.526 | [−0.15 0.14] | 1.76 | −6.33 | <0.001* | 1 | [−0.83 −0.44] | >1000 | −4.51 | <0.001* | 1 | [−0.90 −0.36] | >1000 |
| | wHIP | 71 | −2.83 | 0.012* | 0.946 | [−0.28 −0.05] | 57 | −5.58 | <0.001* | 1 | [−0.43 −0.21] | >1000 | −1.59 | 0.076 | 0.886 | [−0.33 .04] | 6.6 |
| PTSD− | rHIP | 45 | −0.63 | 0.356 | 0.764 | [−0.22 0.12] | 3.6 | −1.78 | 0.041* | 0.998 | [−0.46 0.05] | 76 | −1.09 | 0.168 | 0.943 | [−0.48 0.15] | 5.1 |
| | cHIP | 45 | −3.44 | 0.007* | 1 | [−0.62 −0.17] | 103 | −2.01 | 0.027* | 0.994 | [−0.35 −0.04] | 6.4 | 1.66 | 0.076 | 0.977 | [−0.03 0.47] | 1.1 |
| | PC | 45 | 0.54 | 0.356 | 0.800 | [−0.14 0.28] | 2.8 | −4.55 | <0.001* | 1 | [−0.88 −0.37] | >1000 | −3.62 | 0.001* | 1 | [−1.1 −0.30] | 9.1 |
| | wHIP | 45 | −2.60 | 0.019* | 0.986 | [−0.39 −0.06] | 34 | −2.21 | 0.019* | 0.980 | [−0.37 −0.02] | 97.5 | 0.18 | 0.429 | 0.557 | [−0.21 0.27] | 1.8 |
| PTSD+ | rHIP | 54 | 0.76 | 0.356 | 0.861 | [−0.13 0.30] | 1.1 | −5.44 | <0.001* | 1 | [−0.81 −0.38] | >1000 | −3.62 | 0.001* | 1 | [−1.1 −0.32] | 105 |
| | cHIP | 54 | 0.43 | 0.362 | 0.790 | [−0.16 0.29] | 3.6 | −4.93 | <0.001* | 1 | [−0.66 −0.29] | >1000 | −2.93 | 0.004* | 1 | [−0.89 −0.18] | 3.8 |
| | PC | 54 | 0.57 | 0.356 | 0.810 | [−0.13 0.25] | 1.15 | −6.89 | <0.001* | 1 | [−0.76 −0.43] | >1000 | −4.54 | <0.001* | 1 | [−0.93 −0.36] | >1000 |
| | wHIP | 54 | 0.66 | 0.356 | 0.750 | [−0.12 0.28] | 1.6 | −6.06 | <0.001* | 1 | [−0.72 −0.38] | >1000 | −3.61 | 0.001* | 0.999 | [−0.94 −0.30] | 88 |

We first explored the negative modulation of PE+ (reactive control) and beliefs (predictive control) during top-down suppression. We then compared PE+ and beliefs coupling parameters within each group, to gauge the imbalance between reactive and predictive control (right part of the table). For both analyses, we also report one-tailed t statistics, p values adjusted for false discovery rate correction ($p_{FDR}$; see the "Methods" section), the posterior probability (Pp) of group mean being different from zero, the bootstrapped 95% confidence intervals (CIs) of the mean, and the Bayes factor (BF). df degrees of freedom, rHIP rostral hippocampus, cHIP caudal hippocampus, PC precuneus, wHIP whole hippocampus. *Indicates significant effect as revealed by one-tailed t-tests at $p_{FDR} < 0.05$.

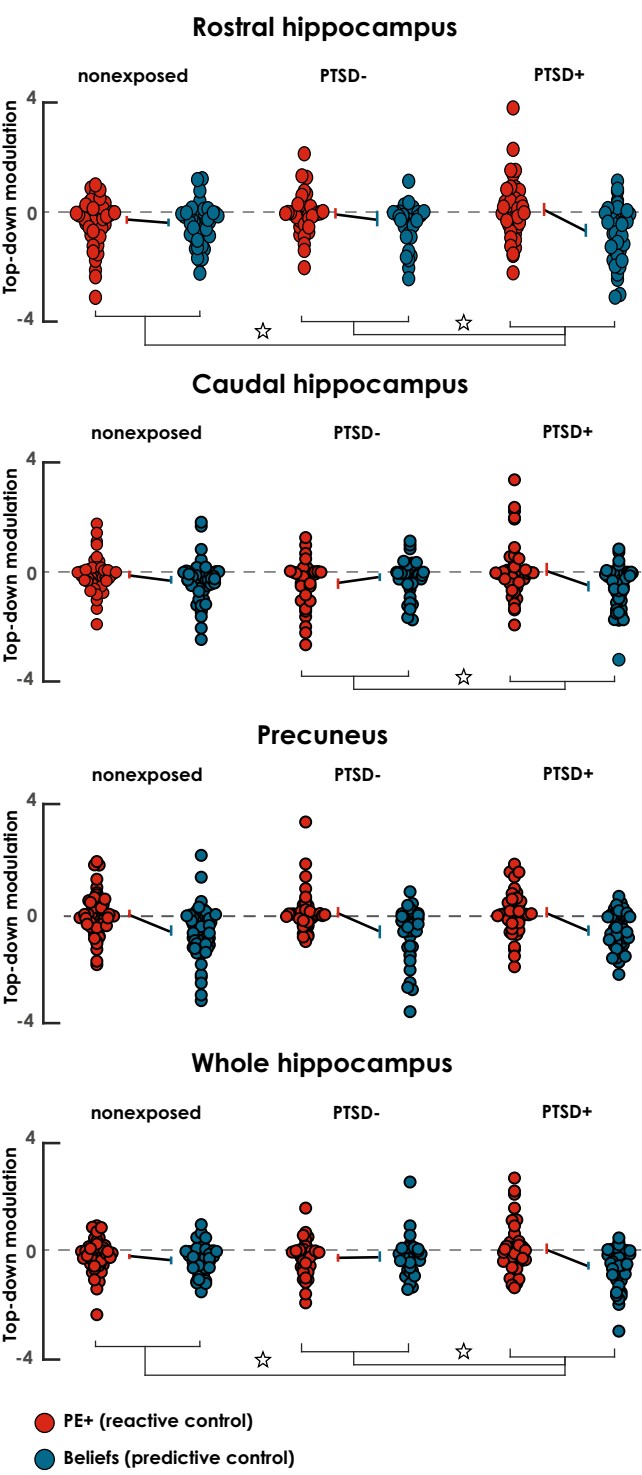

**Fig. 4 BMA of top-down coupling parameters during belief- and PE-driven suppression.** Red and blue circles represent the modulation of PE+ and beliefs on the top-down coupling between the MFG and the target regions, respectively, in non-exposed (n sample size = 72), PTSD− (n sample size = 46) and PTSD+ (n sample size = 55). Lines represent group average coupling parameters ± bootstrapped 95% CI of the group mean, and small circles represent individual participant coupling parameters. '*' indicates significant interaction between groups and the balance in belief-driven predictive and PE-driven reactive control as shown by one-tailed t-tests at $p_{FDR} < 0.05$. See Table 1 for statistical details.

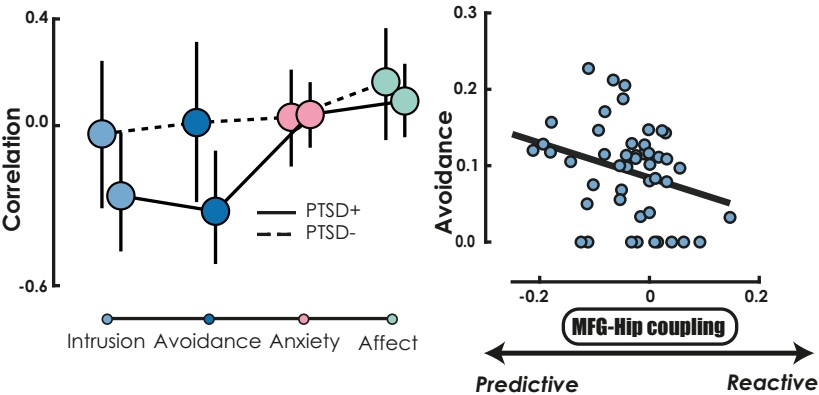

**Fig. 5 Correlation between control imbalance and PTSD symptoms.** Left panel: correlations between the balance of memory control over the hippocampus (i.e., predictive–reactive control coupling parameters) and mental health features for PTSD+ (solid line) and PTSD− (dashed line). The large circles represent the Spearman correlation coefficients and the error bars represents 95% bootstrapped CI of the correlation (and thus indicate significance when they do not overlap with zero). Anxiety-related transdiagnostic features included anxious arousal, dysphoric arousal, and general anxiety, while affect-related transdiagnostic features included anhedonia, mood, and depression. Right panel: relationship between control imbalance in the whole hippocampus and avoidance symptom severity (adjusted for total symptom severity). $N$ sample size = 55 participants with PTSD.

traumatic memory, other symptoms associated with PTSD cross-diagnostic boundaries. A recent study[28] examining trauma, anxiety and mood disorders found three transdiagnostic anxiety-related dimensions: anxious arousal, dysphoric arousal (i.e. tension), and general anxiety, and three transdiagnostic affect-related dimensions: anhedonia, mood and depression. We investigated the relationship between re-experiencing, avoidance, anxiety-related dimension, and affect-related dimension on one hand, and the imbalance of memory control mechanisms regulating the hippocampal activity on the other hand, in both the PTSD+ and PTSD− groups. We tested the hypotheses that excessive predictive control in the PTSD+ group was related to an increase in avoidance and intrusion, and that such negative relationship was significantly stronger than the relationship observed for anxiety- or affect-related dimensions, or the relationship observed in the same dimension but in the PTSD− group. Intrusion, avoidance, mood, anhedonia, dysphoric arousal, and anxious arousal symptoms were obtained from the PTSD checklist for DSM-5 (PCL-5)[29] and were adjusted for total symptom severity to ensure that the correlation with these dimensions were not confounded with PTSD severity. Depression and general anxiety dimensions were obtained using the Beck Depression Inventory and State Anxiety Inventory, respectively. After computing correlation between control imbalance in the wHIP and each of these symptoms, dysphoric arousal, anxious arousal, and general anxiety were summarized to reflect an anxiety-related dimension, while anhedonia, mood, and depression were summarized to reflect affect-related dimension.

In the PTSD+ group, we found that excessive predictive memory control significantly correlated with higher severity of avoidance ($R_{spearman} = -0.32$; 95% CI = [−0.52 −0.09]; $Z$-val = 2.27; $p_{FDR} = 0.047$) and marginally to intrusion symptoms after FDR correction ($R_{spearman} = -0.26$; 95% CI = [−0.47 −0.03]; $Z$-val = 1.84; $p_{FDR} = 0.065$). On the opposite, there was no significant relationship with the severity of both anxiety-related ($R_{spearman} = 0.04$; 95% CI = [−0.08 0.16]; $Z$-val = 0.55; $p_{FDR} = 0.30$) and affect-related ($R_{spearman} = 0.09$; 95% CI = [−0.04 0.23]; $Z$-val = 1.09; $p_{FDR} = 0.18$) transdiagnostic symptoms (see Fig. 5). Crucially, we statistically compared the relationship that predictive control entertains with avoidance and intrusion in the PTSD+ group, to those entertain with trans-diagnostic symptoms (anxiety-related and affect-related dimensions). We used a boostrapping approach to obtain the confidence interval of the correlation difference and the p-value, respectively. Excessive predictive control was

significantly more strongly related to re-experiencing symptoms than with anxiety-related (correlation difference 90% CI [−0.62, −0.14], $Z$-val = 3.09; $p_{FDR} = 0.004$) or affect-related (correlation difference 90% CI [−0.49, −0.04], $Z$-val = 2.36; $p_{FDR} = 0.018$) transdiagnostic clinical features. A similar pattern was observed for avoidance compared with anxiety-related (correlation difference 90% CI [−0.66, −0.22], $Z$-val = 3.6; $p_{FDR} = 0.001$) or affect-related (correlation difference 90% CI [−0.52, −0.13], $Z$-val = 2.82; $p_{FDR} = 0.006$) dimensions. Furthermore, excessive predictive control was significantly more strongly related to avoidance symptoms (correlation difference 90% CI [−0.63, −0.04], $Z$-val = 2.12; $p = 0.034$) in the PTSD+ than in the PTSD− group, although such difference in correlation between groups was not observed for re-experiencing symptoms (correlation difference 90% CI [−0.51, 0.05], $Z$-val = 1.52; $p = 0.13$).

*Imbalance between predictive and reactive control in PTSD reflects independent processes.* Taken together, these findings suggest that individuals with PTSD cannot harmoniously balance predictive and reactive control in the hippocampus, unlike individuals without PTSD. This imbalance might reflect exaggerated predictive control applied in anticipation that prevents the deployment of reactive control. Contradicting this idea, however, predictive regulation of the hippocampus in PTSD+ was not related to reactive control ($R_{spearman} = 0.01$, 95% bootstrapped CI [−0.26, 0.30]).

Alternatively, despite serving the same down-regulation function of memory processes, predictive and reactive control can be conceptualized as two independent, yet downward forces, jointly mitigating hippocampal activity. These two directional forces can be projected on two distinct orthogonal axes (i.e., separated by a 90° angle) in a two-dimensional circular space (see Fig. 6, on the left). In this framework, the imbalance is reflected in the direction of the resultant vector combining the two forces. We fixed the 0° position at the bottom of the y-axis, and computed the direction of the resultant vector with respect to this optimally balanced position (see the "Methods" section). The angle of the resultant vector reflected an imbalance in favor of either predictive control (from 0° to 180°, moving anticlockwise) or reactive control (from 0° to −180°, moving clockwise).

In the hippocampus, we found a significant imbalance in favor of predictive control in the PTSD+ group ($M = 33.35°$; 95% CI [20.2°, 46.2°]) and the nonexposed group ($M = 15.33°$, 95% CI [4.55°, 26.51°]), but not in the PTSD− group ($M = 6.86°$; 95% CI [−9.17°, 23.8; see Fig. 6). When we compared the groups using circular

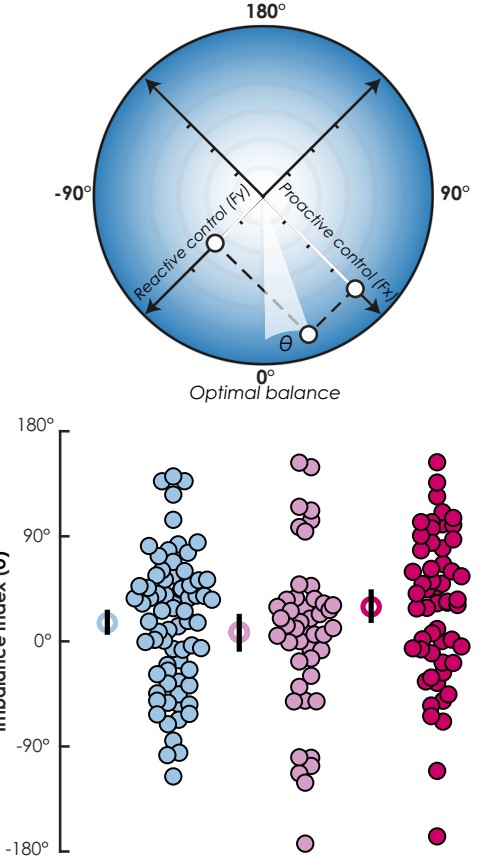

**Fig. 6 Geometric interpretation of predictive and reactive control as perpendicular forces.** Left panel: circular projection of the resultant vector of predictive (i.e., belief) and reactive (i.e., PE+) control forces. 0° represents the optimum balance between these two orthogonal forces. The angle of the resultant force indicates imbalance toward either predictive ($\theta > 0°$) or reactive ($\theta < 0°$) control. The right panel shows the distribution of this imbalance index for each of the three groups. The empty circles represent the group circular average ± bootstrapped 95% CI for nonexposed (*n* sample size = 72), PTSD− (*n* sample size = 46) and PTSD+ (*n* sample size = 55).

statistics[30] (see the "Methods" section), we observed that the imbalance toward predictive control in the hippocampus increased significantly for PTSD+ compared with both the PTSD−, $t(99) = 2.10$, $p = 0.018$, 95% CI [−46.8°, −4.2°], and nonexposed, $t(125) = 1.74$, $p = 0.042$, 95% CI [−35.77°, −0.86°], groups. No differences were found between the nonexposed and PTSD− groups, $t(114) = 0.72$, $p = 0.235$, 95% CI = [−11.3°, 27.86°].

## Discussion

To explain the persistence of intrusive traumatic memories and their avoidance, previous accounts of PTSD have largely focused on the disruption of memory functions[7,31]. More recently, brain connectivity analyses in individuals with PTSD during a memory suppression task revealed a lack of adaptive modulation of top-down control over memory processing in response to intrusive memory cues, suggesting that this persistence may additionally be rooted in the disruption of inhibitory control processes supporting active forgetting[10]. However, the origin of these deficits in top-down control over memory processing remains unknown, and standard analyses of connectivity mask the hidden influence of predictive processing over control processes. Here, we suggest that altered predictive processing[6] constitute a unifying framework that links these two seemingly unrelated accounts of PTSD.

We showed that prediction of future memory control demand related to intrusions drives the flexible adaptation of memory suppression. These dynamic adjustments are orchestrated by a top-down inhibitory signal originating from the right DLPFC, which optimally balanced the suppression of the beliefs of future intrusive re-experiencing and their actual online emergence. This balancing is compromised in individuals with PTSD, but not in resilient or nonexposed individuals. We found that the disproportionate predictive inhibitory control over hippocampal activity based on beliefs, coupled with the reduction in reactive control based on PE+, was specifically related to cardinal features of PTSD related to the trauma, including avoidance and traumatic re-experiencing. This finding echoes recent proposals suggesting that disturbances of predictive processing about threat are central to the expression of PTSD, including avoidance behaviors and traumatic re-experiencing[3,17]. Our findings show that in PTSD, computations conferring higher value on predictions and beliefs than on outcomes also impair control processes, suggesting that maladaptive avoidance responses generalize to memory processes and nonthreatening situations.

Do these observations reflect a genuine, distinct deficit of reactive control in PTSD? The presence of a crossover interaction between control conditions and groups does not guarantee the existence of independent mental processes[32]. The disruption of reactive control may arise from exaggerated predictive control, and not reflect a genuine deficit in the online purging of intrusive memories. Extreme anticipation may prevent the control system from flexibly and adaptively adjusting its response when predictive attempts have failed, suggesting instead a single processing continuum between two modes[33]. This means that there may not necessarily be a second disrupted reactive control mechanism independent of predictive control. This hypothesis, however, seems unlikely, as we did not observe a negative relationship between the magnitudes of predictive and reactive control. Furthermore, we observed an imbalance after treating these two components of control as orthogonal yet downward forces originating from the same point of application (Fig. 6). This illustrates how a single control system could regulate two distinct computational quantities that are independently in the service of the same function (i.e., suppression of unwanted memories). However, these complementary processes take place within the same neurobiological system, which raises the question of how one (predictive control) may be enhanced (or at least preserved) when the other one (reactive control) is disrupted.

The ability to countermand the PE associated with intrusive memories may depend on the availability of executive control resources. Executive resources may be diminished in PTSD following gray-matter atrophy in the right DLFPC[34], or affected by disruption of the white-matter tracts originating from the prefrontal cortex[35]. PE increases attentional demand during learning in individuals with PTSD[4]. Thus, although limited executive functioning may allow for sustained predictive control in the background[36], it may proscribe the more demanding transient regulation of PE associated with intrusive memories. We did not observe any difference between the aMFG and pMFG with respect to predictive or reactive control, suggesting a general disruption of inhibitory executive functions. This finding fits observations in the motor domain suggesting that both forms of control are coordinated and interact in the DLPFC[37].

Alternatively, the current findings may not reflect difficulties of the executive system, but alterations of the receptor system, which converts excitatory projections from the prefrontal cortex into local feedforward inhibition via GABAergic interneurons. It has been suggested that predictive and reactive forms of inhibitory control of the hippocampus are implemented via two distinct neuroanatomical pathways[14]. According to this model, predictive

control processes may preferentially modulate the activity of rhinal inhibitory interneurons to gate inputs to the hippocampus, preventing the initiation of the retrieval process. The extent and nature of rhinal alterations in PTSD remain unclear, compared with alterations of the hippocampus proper. Reactive control, however, may activate CA1 inhibitory interneurons via the thalamic reuniens, a hippocampal subregion particularly involved in the regulation of pattern completion during memory retrieval[14]. Interestingly, studies conducted in rodents suggest that chronic stress affects GABAergic interneurons[38]. These are neurotransmitters that mediate memory control mechanisms in the hippocampus[39] and regulate the activity of dopamine PE neurons[40]. Alteration of this inhibitory function might therefore explain the excessive pattern completion and the lack of control over intrusive memories in individuals with PTSD.

We do not yet know whether the mechanisms identified here are related to the formation and persistence of traumatic memory traces in individuals with PTSD. Proactive avoidance of memories intrinsically implies the preservation of the related memory trace, maintaining the negative beliefs[41]. Furthermore, monitoring of the to-be-avoided representations increases paradoxical rebounds and the persistence of trauma-related memories[42]. Lastly, excessive interruption of hippocampal processing through predictive control may prompt the forgetting of safe contexts[43] associated with trauma reminders and contribute to the overgeneralization of fear. Previous TNT studies in healthy individuals have suggested that motivated forgetting is preferentially linked to the control of intrusive memories crossing the proactive gate[33]. Further investigations are required to evaluate whether the persistence of traumatic memory could be related to an inability to reactively countermand the neural activity associated with PE and involuntarily recall. On the one hand, PE increases the malleability of the memory trace[44] and its control might facilitate forgetting by promoting memory destabilization during the (re) consolidation mechanisms occurring during memory recall[45,46]. On the other hand, predictive coding models of the brain propose that memory recall arises from the disinhibition of pyramidal cells encoding the bottom-up PE[47]. Such disinhibition is orchestrated by the hippocampus and its suppression might increase the plasticity of inhibitory engram and the silencing of neocortical traces[48].

Previous studies defined reactive control based solely on the presence of intrusive memories, without disentangling the confounding influence of predictive control dynamics, possibly leading to misinterpretations of the meaning of inhibitory control observed during memory intrusions. The absence of between-group differences with respect to the PC, previously associated with the suppression of intrusive memories in trauma-exposed individuals without PTSD[10], further illustrates this point. Our neurocomputational approach overcomes this overlap and provides a partial answer to the longstanding question about the relationship between avoidance and memory suppression in PTSD.

Most of the recommended therapeutic treatments that have been shown to be effective for PTSD involve overcoming avoidance of the traumatic experience. Our findings suggest that this avoidance may result from the general disruption of hidden predictive operations engaged to infer and anticipate intrusive memories, biasing their control. Although our findings suggest that such bias is specifically related to trauma-related dimension of PTSD, and not to other transdiagnostic features related to affect or anxiety disorder, future studies would be needed to demonstrate the link between the development of a predictive control disorder and the development of the traumatic memory. Yet, this opens up possible new avenues for understanding the formation and maintenance of the traumatic engram in terms of

predictive control disorder. New interventions designed to modulate and update the traumatic engram after it has been re-indexed in the hippocampus[31] should aim to restore the balance between predictive and error-driven control.

## Methods

**Participants**. Seventy-three non-exposed and one hundred and two exposed subjects participated in this study[10]. Exposed participants were recruited through a transdisciplinary and longitudinal research "Program 13-Novembre" (http://www. memoire13novembre.fr/), a nationwide funded program in partnership with victims' associations. In the current study, the data of two participants (one non-exposed and one exposed) were excluded from the final analyses, as they had an unusually low number of remaining pairs, making it impossible for us to calculate an item-specific belief computational model (see below). The final sample consists of 101 exposed and 72 nonexposed participants. Non-exposed participants were not present in Paris on 13 November 2015 and were recruited from local panel of volunteers. All the participants are between the ages of 18 and 60 years old, right-handed, French speaking and had a body mass index inferior to 35 kg/m². A clinical interview with a medical doctor was conducted to ensure that participants had no reported history of neurological, medical, visual, memory, psychiatric disorders. Exclusion criteria also included history of alcohol or substance abuse (other than nicotine), mental or physical condition that preclude MRI scanning (e.g., claustrophobia or metal implants) and medical treatment that may affect the central nervous system or cognitive functions.

Exposed participants were diagnosed using the Structured Clinical Interview for DSM-5 (SCID)[49] conducted by a trained psychologist and supervised by a psychiatrist. All the exposed participants met DSM-5 criterion A indicating that they experienced a traumatic event. Exposed participants were diagnosed with PTSD in its full form if all the additional diagnostic criteria defined by the DSM-5 were met ($n = 29$). Participants were diagnosed with PTSD in its partial form ($n = 26$) if they had re-experiencing symptoms (criterion B), with persistence of the symptoms superior to one month (criterion F) that caused significant distress and functional impairment (criterion G)[10,50]. Trauma-exposed participants with full and partial PTSD profiles were grouped together for the purpose of statistical analyses in one unique clinical group referred to as the PTSD group[10]. The study includes 55 trauma-exposed participants with PTSD (PTSD+, 30 females and 25 males, mean age = 37.14, SD = 8.35), 46 trauma-exposed participants without PTSD (PTSD−, 16 females and 30 males, mean age = 36.84 years, SD = 7.05 years) and 72 nonexposed participants (38 females and 34 males, age = 33.69 years, 33.66, SD = 11.40 years).

PTSD symptoms severity was assessed with the Post-traumatic Stress Disorder Checklist for DSM-5 (PCL-5)[51]. To assess for anxiety and depression, State-Trait Anxiety Inventory (STAI)[52] and Beck Depression Inventory (BDI)[53] were also administered. All the participants completed the study between 13 June 2016 and 7 June 2017. Participants were financially compensated for their participation in the study. The study was approved by the regional research ethics committee ("Comité de Protection des Personnes Nord-Ouest III", sponsor ID: C16-13, RCB ID: 2016-A00661-50, clinicaltrial.gov registration number: NCT02810197). All the participants gave written informed consent before participation, in agreement with French ethical guidelines. Participants were asked not to consume psychostimulants, drugs, or alcohol prior to or during the experimental period.

**Materials**. The stimuli were three series of lists of 72 word–object pairs composed of neutral abstract French words[54] and objects selected from the Bank Of Standardized Stimuli (BOSS)[55]. Three series of four lists of 18 pairs assigned to four conditions (think, no-think, baseline, and unprimed for the final priming test task after the Think/No-think phase) were created, plus 8 fillers used for practice. The lists of pairs were presented in counterbalanced order across the three series, the four conditions and the three groups of participants and matched on different properties that may influence performance to the task. The lists of words were matched on average naming latency, number of letters and lexical frequency[54]. The lists of objects were matched relative to the naming latency, familiarity and visual complexity levels, viewpoint, name and object agreement and manipulability[55]. Stimuli were presented using the Psychophysics Toolbox implemented in MATLAB (MathWorks). We used neutral material completely disconnected from the traumatic experience that enables to investigate general memory control mechanisms and incidentally avoid ethical issues for the trauma-exposed group.

**Procedure**. Before MRI acquisition, participants learned 54 French neutral word-object pairs that were presented 5 s each. After the presentation of all pairs, the word cue for a given pair was presented on the screen for up to 4 s and participants were asked whether they could recall and fully visualize the paired object. If so, three objects then appeared on the screen (one correct and two foils), and participants had up to 4 s to select which object was associated with the word cue. After each recognition test, the object correctly associated with the word appeared 2500 ms on the screen and participants were asked to use this feedback to increase their knowledge of the pair. Pairs were learned through this test–feedback cycle procedure until either the learning criterion (at least 90% of correct responses) was reached or a maximum of six presentations was achieved. Once participants had

reached the learning criterion, their memory was assessed for one last time using a final criterion test on all of the pairs but without giving any feedback on the response. No group differences were found on this final criterion test (all $Ps > 0.18$). Following this learning phase, pairs were divided into 3 lists of 18 pairs assigned to think, no-think, and baseline conditions for the Think/No-think task (TNT). Participants were given the think/no-think phase instructions and a short TNT practice session before MRI acquisition to familiarize them to the task.

Following this TNT practice session, participants entered the MRI scanner. During the T1 structural acquisition, the complete list of learned pairs was presented once again to reinforce the learning of the pairs (5 s for each pair). This overtraining procedure was intended to ensure that the word cue would automatically bring back the associated object. Following this reminder of the pairs, participants performed the TNT task, which was divided into four sessions of about 8 min each. In each session, the 18 think and 18 no-think items were presented twice. Word cues appeared for 3 s on the screen and were written either in green for think trials or in red for no-think trials. During the TNT practice session, participants were trained to use a direct suppression strategy. During the no-think trials, participants were instructed to imperatively prevent the object from coming to mind and to fixate and concentrate on the word-cue without looking away. Participants were asked to block thoughts of the object by blanking their mind and not by replacing the object with any other thoughts or mental images. If the object image came to mind anyway, they were asked to push it out of mind. After the end of each think or no-think trial cues, participants reported whether the associated object had entered awareness by pressing one of two buttons corresponding to "yes" (i.e., even if the associated object pops very briefly into their mind) or "no". Although participants had up to 3600 ms to make this intrusion rating, they were instructed to make it quickly without thinking and dwelling too much about the associated object. The rating instruction was presented up to 1 s on the screen and followed by a jittered fixation cross (1400, 1800, 2000, 2200 or 2600 ms). The Genetic Algorithm toolbox[56] was used to optimize the efficiency of the Think versus No-Think contrast. 20% additional null events with no duration and followed by the jittered fixation cross only were added.

Finally, during a debriefing questionnaire, participants were asked about the strategies used during the TNT phase. Participants rated on a 5-point scale (0: never; 4: all the time) the degree to which they used different kind of strategies to prevent the object from coming to mind during the No-Think condition (i.e., direct suppression, thought substitution or another strategy). This questionnaire was administered to determine whether participants complied with the direct suppression instructions. Debriefing confirmed that the participants remained attentive to the word displayed on the screen and predominantly controlled the unwanted memories by directly suppressing the associated object. Participants engaged significantly less in other strategies than in direct suppression to control awareness of the No-think items (Wilcoxon's signed Rank test: $z > 140$, $p < 0.001$). Moreover, Kruskal–Wallis tests did not evidence any difference between the groups for any kind of strategies used ($H(2) < 2.73$, $ps > 0.26$).

**MRI acquisition parameters**. MRI data were acquired on a 3 T Achieva scanner (Philips). All participants first underwent a high-resolution T1-weighted anatomical volume imaging using a 3D fast field echo (FFE) sequence (3D-T1-FFE sagittal; TR = 20 ms, TE = 4.6 ms, flip angle = 10°, SENSE factor = 2, 180 slices, $1 \times 1 \times 1$ mm$^3$ voxels, no gap, FoV = $256 \times 256 \times 180$ mm$^3$, matrix = $256 \times 130 \times 180$). This acquisition was followed by the TNT functional sessions which were acquired using an ascending T2-star EPI sequence (MS-T2-star-FFE-EPI axial; TR = 2050 ms, TE = 30 ms, flip angle = 78°, 32 slices, slice thickness = 3 mm, 0.75 mm gap, matrix $64 \times 63 \times 32$, FoV = $192 \times 192 \times 119$ mm$^3$, 235 volumes per run). Each of the 4 TNT functional sequence lasted about 8 min.

**fMRI preprocessing**. Images preprocessing were first conducted with the Statistical Parametric Mapping toolbox (SPM 12, University College London, London, UK). Functional images collected during the TNT phase were (1) spatially realigned to correct for motion (using a 6-parameter rigid body transformation); (2) corrected for slice acquisition temporal delay; and (3) coregistered with the skull-stripped structural T1 image. The T1 image was bias-corrected and segmented using tissue probability maps for gray matter, white matter and cerebrospinal fluid. The forward deformation field (y_*.nii) was derived from the nonlinear normalization of individual gray matter T1 images to the T1 template of the Montreal Neurological Institute (MNI). Each point in this deformation field is a mapping between MNI standard space to native-space coordinates in mm. Thus, this mapping was used to project the coordinates of the MNI standard space ROIs to the native space functional images.

**Computational modeling**. We used computational modeling to investigate participants' beliefs about upcoming intrusive memories in the no-think condition of the TNT task. Taking the *observing the observer* meta-Bayesian approach[19] one step further, our aim was to *observe the observer observing him- or herself*. According to this approach, agents use a perceptual model to make inferences about the hidden states that control the world. The observation (or response) model describes the relationship between inferred hidden states and behavioral

outcomes. In our models, inputs $u$ and outcomes $y$ were binary:

$$u^{(t)} \in \{0, 1\}; y^{(t)} \in \{0, 1\} \tag{1}$$

where 0 corresponds to nonintrusion and 1 to intrusion at time $t$. As our aim was to model participants' beliefs about their own intrusion ratings during the TNT, input $u$ at time $t$ was outcome $y$ at time $t-1$:

$$u^{(t)} = y^{(t-1)} \tag{2}$$

To model individual time series of internal beliefs, we used the HGF, KF, and RW models implemented in the TAPAS toolbox (available at https://www.tnu.ethz.ch/de/software/tapas), which applies variational Bayesian inversion to infer hidden states maximizing the log-model evidence (LME).

*Perceptual models*
Two-level hierarchical Gaussian filter: We used a two-level HGF as a perceptual model. Developed by Mathys et al.[23], the HGF assumes that agents form internal beliefs in a hierarchical fashion. Implementing a variational approximation approach, the HGF allowed us to estimate trial-by-trial trajectories of internal beliefs at multiple levels. The lowest level corresponds to participants' beliefs about whether they are experiencing a memory intrusion or not ($x_1$). As $u^{(t)}$ and $y^{(t)}$ are binomial, $x_1$ assumes a Bernoulli distribution. Accordingly, first-level beliefs $x_1$ are the logistic sigmoid transformations of second-level beliefs $x_2$ which, by contrast, are unbounded:

$$x_1^{(t)} \sim \text{Bernoulli}\left(\frac{1}{(1 + \exp(-x_2^{(t)}))}\right) \tag{3}$$

The second level ($x_2$) corresponds to participants' internal beliefs about the volatility of memory intrusions experienced during the TNT task: $x_2$ is denoted as a Gaussian random walk whose step size is controlled by the free parameter $\omega$. The resulting beliefs assume Gaussian distributions described by their sufficient statistics: posterior mean $\mu$ and uncertainty $\sigma$ (i.e., variance):

$$x_2^{(t)} \sim \text{N}(x_2^{(t-1)}, \exp(\omega)) \tag{4}$$

where the $\omega$ parameter controls the variance of $x_2$, shaping the magnitude at which beliefs are updated. We used the superscript ^ to indicate prior internal beliefs. For example, $\mu^{(t)}$ represents posterior internal beliefs at Trial $t$, and $\hat{\mu}^{(t)}$ represents internal beliefs prior to the outcome $y^{(t)}$ (intrusion or nonintrusion).

The variational approximation underlying the HGF model fitting allowed participant-specific free parameters to be estimated, along with the trial-by-trial trajectories of internal belief updating, which were determined by the participants' sets of parameters. Crucially, the updating of second-level beliefs $\mu_2^{(t+1)} - \mu_2^{(t)}$ in the model is proportional to ascending first-level prediction errors weighted by their uncertainty:

$$\mu_2^{(t+1)} - \mu_2^{(t)} \propto \Psi^{(t)}\delta^{(t)} \tag{5}$$

where $\Psi$ is a weighting factor representing the inverse of second-level belief precision $\pi_2^{(t)}$ (i.e., uncertainty):

$$\Psi^{(t)} = \frac{1}{\pi_2^{(t)}} \tag{6}$$

This quantity is modulated by the $\omega$ parameter, and $\delta$ represents PE, namely the difference between beliefs after and before presentation of a stimulus:

$$\delta^{(t)} = \mu_1^{(t)} - \mu_1^{(t-1)} \tag{7}$$

As participants were instructed to report whether or not they experienced a memory intrusion at time $t$, posterior beliefs are equal to the outcome:

$$\mu_1^{(t)} = y^{(t)}$$

The belief updating equation allowed us to estimate participants' predictions $\hat{\mu}^{(t)}$ about the outcome $y^{(t)}$ before it occurred. Importantly, as $\mu^{(t-1)}$ corresponds to prior internal beliefs about the outcome (i.e., sigmoid transformation of $\mu_2$),

$$\hat{\mu}_1^{(t)} = \frac{1}{1 + \exp(-\mu_2^{(t-1)})} \tag{8}$$

PE (or $\delta^{(t)}$) represents the divergence between the real outcome (i.e., intrusion/nonintrusion) and the predicted one:

$$\text{PE}^{(t)} = y^{(t)} - \hat{\mu}_1^{(t)} \tag{9}$$

Next, according to Eq. (5), the updating of posterior beliefs about the tendency to experience intrusions (i.e., $\mu_2^{(t)}$) is driven by the quantification of prediction failure (i.e., $\text{PE}^{(t)}$), weighted by uncertainty about the beliefs (i.e., $\frac{1}{\pi_2^{(t)}}$ in Eq. (6)). Thus, when beliefs are more uncertain, PE has a greater impact on belief updating, improving future predictions about upcoming trials. Importantly, by shaping the uncertainty of beliefs, $\omega$ plays a crucial role in their updating.

For model fitting, we used prior parameters defined in de Berker et al.[57], who conferred high variance on $\omega$ priors (mean: −3, variance: 16) in order to efficiently

catch any possible between-participants variability on this parameter. It should be noted that 10 of the 173 participants included in this study showed no modification of the $\omega$ parameter in its prior state (i.e., −3; see Fig. 2e). This absence of departure from the prior mean was due to the presence of a stochastic occurrence of intrusion rating (with a mean consistently close to 0.5 throughout the task), prohibiting the consistent updating of this parameter. It should, however, be noted that the belief trajectories were still valid for these participants and could be used to infer model accuracy or in subsequent connectivity analyses.

Kalman filter: To include the hypothesis that internal beliefs about experiencing intrusions are uncertain, dynamically updated, but nonvolatile (contrary to HGF), we included a KF[22,58] in our model space. Like the HGF, the KF estimates the trial-by-trial weighting of PE in belief updating, but in this model, beliefs are not hierarchical, and uncertainty therefore remains constant during learning. In the KF framework, beliefs about experiencing an intrusion $\hat{\mu}$ are updated as follows:

$$\hat{\mu}^t = \hat{\mu}^{(t-1)} + K\delta^{(t-1)} \qquad (10)$$

where $K$ is the Kalman gain, representing trial-by-trial learning. The gain is modulated by two free parameters ($\pi$ and $\omega$) that encode belief reliability and uncertainty:

$$K^t = \frac{K^{(t-1)} + \pi\omega}{K^{(t-1)} + \pi\omega + 1} \qquad (11)$$

These two free parameters model two different aspects of belief updating: $\pi$ quantifies how far beliefs can be trusted, based on previous trial history, and $\omega$ quantifies the process variance (i.e., how uncertain the beliefs are).

Rescorla-Wagner: To include the hypothesis about the role of trial-by-trial weighting of PE during intrusion control, we compared the HGF2 and KF models with a traditional reinforcement learning model: the RW[21]. Briefly, RW, HGF and KF share a similar general update equation[23], defined by a weighting factor and prediction error. However, RW assumes a participant-specific fixed learning rate $\alpha$:

$$V^{(t)} = V^{(t-1)} + \alpha(\lambda - V^{(t-1)}) \qquad (12)$$

where $V$ is the prediction and $(\lambda - V^{(t-1)})$ the prediction error (i.e., divergence between real outcome $\lambda$ and prediction at previous trial).

*Source models.* Perceptual models were built using intrusion ratings either from the entire sequence of trials (state model), or separately for each pair of word–object memories (item model), including eight repetitions in total. After concatenation of item trajectories, state and item belief trajectories were linked to an observation model either separately or in combination. Observation models linked the inferred hidden states to the outcomes, describing the probability of observing an outcome $y$ given model parameters. For each model in the perceptual model space, we built an observation model based on beta density probability distributions:

$$p(y|\theta) = \frac{\Gamma(\alpha+\beta)}{\Gamma(\alpha)\Gamma(\beta)}y^{(\alpha-1)}(1-y)^{(\beta-1)} \qquad (13)$$

where $\theta$ refers to participants' beliefs estimated through the different perceptual models, $\Gamma$ expresses a Gamma function, $\alpha = \theta * \nu$, and $\beta = \nu - \alpha$, $\nu$ is a participant-specific free parameter (i.e., inverse decision noise regulating beta density width, estimated during model fit). Here, the observation model described the accuracy of internal beliefs about outcomes (i.e., intrusions). Note here that the beta observation model performed better than other observation function such as the softmax response model (because the log-probability of choice of the beta observation model does not change sharply around belief equal to 0.5, preserving model accuracy). However, although our data do not involve such extreme cases, this model contains the slight absurdity that when beliefs approach certainty (i.e. near 1 or 0), the corresponding probability of choice starts to sink again. For all three models in the perceptual model space (i.e. RW, KF, and HGF), we built the following three source models.

- The *state* source model hypothesized that belief $\theta$ ($\hat{\mu}_{1s}$ for HGF, $\hat{\mu}_s$ for KF and $V_s$ for RW) at trial $t$ was influenced by previous trial history, irrespective of the content of the specific item.
- For the *item* source model, we extracted beliefs for each specific no-think item. Throughout the TNT, up to 18 different items (i.e., objec–word pairs) were repeated (on average, $16.29 \pm 2.18$, no group differences, after accounting for error or absence of criterion recall test before trial phase). For each item $i$, we estimated the trajectories of participants' predictions based exclusively on the item's specific history. For these item-specific models, $t$ in Eqs. (1)–(15) refers to the number of times the item $i$ was repeated, instead of the overall no-think trial count. The trajectory of item-based beliefs is referred to as $\hat{\mu}_{1i}$ for HGF, $\hat{\mu}_i$ for KF, and $V_i$ for RW. After estimations, these separated item-based beliefs were concatenated to form a single trajectory.
- In the *combined* source model, we considered a scenario in which participants combined state and item beliefs to improve prediction accuracy. A joint posterior distribution with mean $\hat{\mu}_c$ was created (starting from the second repetition of each item) by summing the two types of

beliefs, weighted for their respective accuracy, and dividing the result by the sum of the variances:

$$\theta = \hat{\mu}_c = \frac{\hat{\mu}_{1s}\hat{\pi}_{1s} + \hat{\mu}_{1i}\hat{\pi}_{1i}}{\hat{\pi}_{1s} + \hat{\pi}_{1i}} \qquad (14)$$

This combined model hypothesized that participants lent more weight to the most accurate (i.e., least uncertain) source of beliefs when creating combined beliefs $\hat{\mu}_c$. For the KF and RW models, we averaged $\hat{\mu}_s$ and $\hat{\mu}_i$, and $V_s$ and $V_i$, respectively.

*Model estimation and accuracy.* The final model space therefore included nine models: state-HGF2, item-HGF2, combined-HGF2, state-KF, item-KF, combined-KF, state-RW, item-RW, and combined-RW. Free perceptual parameters and corresponding belief trajectories were estimated using a quasi-Newtonian optimization algorithm[23]. For state, item, combined trajectories of belief, we computed model accuracy using the sum of the negative log-likelihood of the choice probability.

**Validation of computational modeling**

*Model falsification.* A common issue in computational modeling is how to assess the performance of a set of different models in generating plausible data, given that generative and predictive performances of a model can sometimes be dramatically different[24]. This is an important step that allows the models that best generate plausible data to be identified and those with poor generative performances to be rejected. This procedure is known as *model falsification*[24].

The goal of these simulations was to establish whether the models were able to generate the behavioral reduction in intrusion proportion that we observed across the four blocks of the TNT task (see Fig. 2). We designed a virtual experimental setting with 144 suppression cues distributed across 4 TNT sessions, as in our real experiment. We started with a belief of 0.5 for the first trial, and at each new simulated trial, we generated a new belief based on the perceptual model considered and randomly drawn corresponding perceptual parameters. A suppression parameter was introduced to simulate memory suppression and to avoid the tilting of belief trajectories toward 1. This parameter was initially fine-tuned separately for each model using a grid search to minimize the difference between simulated data and real intrusion profile. After applying this suppression factor to the generated belief, and adding some noise, we computed the negative log-model accuracy of previous responses using the beta observation model (i.e. summing all trials response log-probabilities up to the new one), and generate a new response (i.e. *yes* or *no*) depending on log-model accuracy improvement (i.e. we selected the response for this new trial that best improved the overall log-model accuracy). The inverse decision noise parameter ($\nu$; see above) of the beta observation model was fixed to $e^0$ (i.e., 1), allowing the mapping to be unbiased toward a preferred outcome.

We simulated 200 virtual participants using this procedure, and repeated the virtual experiment 100 times using perceptual parameter randomly drawn from a Gaussian prior distribution tailored to match our own data (to sample plausible parameters), resulting in 20,000 simulations for each of the nine computational models. Then, binary rating generated for each of these 200 simulations were averaged across repeated sampling and summarized as intrusion proportion across the four artificial TNT sessions. We tested the relationship with the real intrusion proportions for our cohort by computing both the mean difference (MD) and the mean correlation (MC) between real and simulated intrusion ratings across the 200 virtual participants. While the MD between simulated and fitted parameters is informative of the absolute distance between real and simulated intrusion ratings, MC indicates whether simulated intrusions mimic the decrease in intrusion rating across normally observed across TNT sessions. We found that for HGF2, both state (MD = 0.069 ± 0.02; MC = 0.543 ± 0.05) item (MD = −0.008 ± 0.01; MC = 0.367 ± 0.03) and combined (MD = 0.054 ± 0.02; MC = 0.575 ± 0.05) models were able to generate data both intercepting the session-wise mean intrusion rating and mimicking the decrease in intrusion proportion across the TNT blocks. Concerning the RW models, only the item model generated the expected patterns of intrusions (MD = −0.004 ± 0.01; MC = 0.769 ± 0.03). While both state and combined models simulated intrusions showed acceptable correlations with real intrusion data (state: MC = 0.765 ± 0.03; combined: MC = 0.274 ± 0.05), both failed in intercepting the session-wise mean of the real intrusion data (state: MD = 0.183 ± 0.02; combined: MD = 0.140 ± 0.01). Similarly to RW, also for KF only the item model generated the expected patterns of intrusions (MD = −0.019 ± 0.01; MC = 0.769 ± 0.03), while both state and combined models showed acceptable correlations (state: MC = 889.0 ± 0.01; combined: MC = 0.292 ± 0.01, $p < 0.001$) but not mean differences (state: MD = 0.292 ± 0.01, combined: MD = 0.278 ± 0.01). The main outcomes of this model falsification analysis can be found in Fig. 2.

*Recovery analyses.* Given this evidence for the generative performances of our models, we addressed another possible pitfall in the model selection workflow: the ability of a set of models to recover their trajectories of belief and the associated perceptual parameters. This analysis further tests the generative performance of a model, by verifying whether the fitting procedure produces meaningful trajectories

and/or parameters, namely the true parameters and the corresponding trajectories used to generate the data[25]. We fitted the different models to the synthetic data, in order estimate the trajectories and the free parameters.

For trajectory recovery, we computed the correlation between fitted and simulated trajectories. We then identified the fitted model among competitors that has the maximum correlation with the simulated trajectory (coding 1 for the best model, and 0 otherwise), and averaged these outcomes across simulations. We computed the inversion matrix, to ensure that the beliefs fitted by a given model was the model that most likely has generated those beliefs (i.e. reverse inference; Fig. 2).

When comparing computational models, it is also important to verify the reliability of the model selection criterion for identifying the true generative model within a set of competitive models, and ensure that this selection is not biased in favor of one particular model[24,25]. This procedure, known as model recovery, consists in simulating data with one specific model and then comparing the predictive performances (i.e. model accuracy) of a set of different models using Bayesian inference. For each of the 200 virtual participants, we first summed the model accuracy across the 100 random sampling. We then identified, for each simulated model, the best fitting model associated with the maximum accuracy, and summarized the probability into a confusion matrix to create the corresponding inversion model[25] (see Fig. 2).

For parameter recovery, we computed the correlation between simulated parameter that generated the data, and the corresponding fitted parameters. This correlation was computed for each of the 200 virtual participants, using 100 randomly sampled free parameters (see above), and then averaged across virtual participants. We found that HGF models had the best overall ability to recover the parameter ω, with small correlations for the state model ($r(98) = 0.263 \pm 0.08$, $p = 0.008$) and moderate correlations for the item ($r(98) = 0.395 \pm 0.08$, $p < 0.001$) model. Significant recovery of $\alpha$ was found in RW models for the state ($r(98) = 0.268 \pm 0.13$, $p = 0.007$), but not for item ($r(98) = 0.013 \pm 0.10$, $p = 0.898$) model. No significant correlations were found between simulated and fitted ω (state: $r(98) = 0.008 \pm 0.10$, $p = 0.937$; item: $r(98) = 0.001 \pm 0.09$, $p = 0.992$) and π (state: $r(98) = -0.004 \pm 0.09$, $p = 0.968$; item: $r(98) = -0.007 \pm 0.09$, $p = 0.945$) in KF models (see Fig. 2).

### Computational dynamic causal modeling

*Regions of interest*. DCM entails a priori selection of regions of interest (ROIs). There is evidence for a central role of the right PFC, particularly the MFG, in inhibiting the memory system during motivated forgetting[11,27,59]. The ROIs included in the DCM models were aMFG and pMFG, rHIP and cHIP, and PC. We initially selected the ROIs from the Brainnetome atlas (BNA[60], http://atlas.brainnetome.org/), which is a fine-grained connectivity-based atlas featuring 210 cortical and 36 subcortical cross-validated brain regions, defined in Montreal Neurological Institute (MNI) space. The aMFG region included A46 (center coordinates: $x = 28$, $y = 55$, $z = 17$) and A9/46v ($x = 42$, $y = 44$, $z = 14$), pMFG included A9/46d ($x = 30$, $y = 37$, $z = 36$) and A8vl ($x = 42$, $y = 27$, $z = 39$), rHIP and cHIP corresponded to two ROIs ($x = 22$, $y = -12$, $z = -20$ and $x = 29$, $y = -27$, $z = -10$), and PC corresponded to dmPOS ($x = 16$, $y = -64$, $z = 25$). The MNI coordinates of the five ROIs were projected onto participants' native space using the deformation field, without any spatial warping or smoothing of the functional images, to ensure maximum accuracy. However, for there to be sufficient demarcation between the aMFG and mMFG signals, aMFG coordinates were initially limited to $y > 35$ mm, and pMFG coordinates to $y < 25$ mm.

For the DCM analysis, we summarized the signals for each participant and each of these ROIs from the averaged time series of 30 contiguous voxels (1012.5 mm³) that were the most significantly related to the main task around the maximum activation peak (using no-think > think contrast for aMFG and pMFG, and no-think < think contrast for memory regions)[27]. To this end, a univariate analysis was conducted on the timecourse of each native space ROI for each participant, by implementing a general linear model (GLM) in SPM12. The voxelwise fMRI time series were high-pass filtered, with a cut-off period of 128 s. Task-related regressors were created by convolving a box-car function at the onset of cue words with the canonical hemodynamic response function. Further regressors of no interest included the six realignment parameters to account for motion artefacts, session dummy regressors, and filler item regressors (i.e., no button press, or no recall during the final criterion test or during think trials). fMRI time series autocorrelations were corrected by entering a first-order autoregressive model of temporal autocorrelation of noise and a white-noise model was estimated using restricted maximum likelihood. The data were then adjusted for confounds, filtered, and whitened using the estimated temporal autocorrelation of noise to correct for non-sphericity. Beta parameters for think and no-think conditions were estimated during a second pass of the general linear model with the ordinary least-square method, and used to calculate participant-specific $t$ maps for each ROI.

*Neural and hemodynamic models for DCM*. DCM[61] allows changes in effective connectivity between a set of brain regions to be inferred by creating and comparing different hypothesis-driven generative models of neural dynamics. It relies on the following general bilinear state equation for these dynamics:

$$\frac{dx}{dt} = \left( A + \sum_{j=1}^{m} u_j B^{(j)} \right) x + Cu \qquad (15)$$

Given $m$ known inputs, the hidden neural dynamics ($\frac{dx}{dt}$) are estimated by relating the activity of each region to the activity of other regions, via (1) intrinsic connections in the absence of experimental manipulations (A matrix), (2) $j$th modulatory input $u_j$ operating on intrinsic connections during experimental conditions (B matrix), and (3) extrinsic input driving activity in the network (C matrix). These neural models are then combined with a hemodynamic model describing the mapping of neural activity onto the BOLD response observed on fMRI (i.e., the Balloon model[62]). The neural and hemodynamic model parameters are estimated through variational Bayes under Laplace approximation, which optimizes model evidence by minimizing free energy and ensures Gaussian posteriors[63].

Two modularity input functions operated on intrinsic connections. The first one corresponded to a boxcar function encoding no-think trials onset and duration, and whose height was parametrically modulated by internal beliefs ($\hat{\mu}_c$, see Eq. (14)). The second corresponded to a boxcar function reflecting only intrusive trials, parametrically modulated by PE (see Eq. (9)). This allowed us to investigate how the discrepancies between internal beliefs and intrusive outcomes were reactively processed by the memory control system, our primary interest. PE was therefore only positive here (PE+), meaning that negative and positive coupling parameters could be interpreted as such. It should, however, be noted that the extent and the sign of the posterior coupling parameters were estimated with respect to the implicit baseline (i.e., unmodeled signal). Here, the neural dynamics were only modeled during no-think trials. The implicit baseline included think trials, and the coupling parameters therefore reflected the modulation of coupling with respect to a baseline corresponding to a mixture of rest (i.e., no stimulation) and memory retrieval. Given that our design included few resting periods, this procedure ensured better isolation of inhibitory mechanisms during memory control. The parametric modulators were not orthogonalized, and were extracted from the winning computational model (i.e., combined-HGF2).

*DCM model space*. All the models assumed bidirectional intrinsic connections between all five regions in the A matrix. This was confirmed by a preliminary analysis that only modeled driving inputs[64]. We created 42 DCM models, which could be divided into three families of fourteen models each and a null family containing two models. The first family (computational top-down modulation family; Fig. 3a, top left) hypothesized that the modulation of PE+ and beliefs occurs during top-down coupling originating from the source regions (i.e., aMFG and pMFG) and targeting memory regions (i.e., rHIP, cHIP and PC). This family could be further divided into two subfamilies encoding different hypotheses on the involvement of aMFG and pMFG in either predictive or reactive control. More specifically, the first subfamily contained seven models encoding all the possible pathways from MFGs to target regions (Fig. 3b), hypothesizing that aMFG and pMFG are involved in reactive (i.e., PE+ modulation) and predictive (i.e., beliefs modulation) control, respectively. Importantly, while beliefs were computed before the actual outcome, including therefore in the no-think trials, PE+ only occurred when participants experienced an intrusion. For this reason, in the first subfamily, intrusion inputs entered the aMFG, while no-think inputs entered the pMFG. The opposite scenario was hypothesized in the second subfamily, with the aMFG and pMFG receiving inputs from no-think and intrusion cues, and modulating control of belief and PE+, respectively. The second family (computational bottom-up modulation family; Fig. 3a, right panel) hypothesized that computations modulate the bottom-up connections from target to source regions, with analogous subdivisions regarding the involvement of aMFG and pMFG with respect to belief and PE+ modulation. The third family (no-computation modulation family; Fig. 3a, bottom left panel) contained 14 models including modularity input functions with no further parametric modulation. Finally, a fourth family containing two null models was added to verify the hypotheses that our modulatory parameters did have an impact on connections, compared with models that did not include these additional modulations (Fig. 3a, bottom right).

### Bayesian model selection and averaging

BMS compares different generative models, in order to select the most probable one. This allows competitive hypotheses on the hidden mechanisms that generated the data[65] to be tested. Here, for both computational and DCM model comparisons, we used a random-effect BMS (i.e., assuming that models can differ between participants) and a free energy approximation of the LME, accounting for both the accuracy and complexity of the models[65]. Interestingly, BMS can be used to compare different families of models, where the model space is partitioned into several models sharing some common underlying hypotheses. For DCM BMS analyses, we first computed the log-family evidence, which summarizes the evidence for models belonging to a given family, assuming prior and posterior additivity of model probabilities into family probabilities, as well as a uniform prior within families[66]. We then compared this evidence using random-effect BMS implemented in the VBA toolbox[67]. Besides computing the probability of one model being more likely than the others in the model space (i.e., exceedance probability, XP), the VBA toolbox estimates the probability that potential differences in model frequencies are due to chance (i.e., Bayesian omnibus risk, BOR). XP and BOR can then be used to compute the PXP, which quantifies the probability of one model being more frequent than others in the model space, above and beyond chance[65].

Despite the remarkably high PXP for the whole sample, BMS did not guarantee that the same model was uniformly the best in all three groups. Traditionally,

independent RFX-BMS has been used to establish the winning model in each separate group. However, this approach does not test the hypothesis that the same model optimally describes data in the different groups. To test this hypothesis, we adopted a recent method[65] implemented in the VBA toolbox[67], which allows between-group model comparisons. This technique computes the probability that different groups are sampled from a single population in which the elected model best explains the data.

We performed BMA across the 14 DCM models that belonged to the wining computational top-down modulation family (see Results section). BMA yields posterior coupling parameters specific to each participant that are weighted by participant-specific posteriors. The optimum model within the selected family is treated as a random effect across participants[68]. For each participant $s$ belonging to the group $g$ (i.e., nonexposed, PTSD−, or PTSD+), the averaged parameters across the 14 models of the family, $P(\theta_{s \in g} | Y, m \in f_D)$, are computed by weighting the participant's posteriors for each model $m$ in the family (i.e., $P(\theta_s | y_s, m)$) by the posterior probabilities that participant $s$ uses model $m$ (i.e., $P(m_s | Y_g)$):

$$P\left(\theta_{s \in g} \mid Y_g, m \in f_D\right) = \sum_{m \in f_D} P\left(\theta_{s \in g} \mid y_{s \in g}, m\right) P\left(m_{s \in g} \mid Y_g\right) \qquad (16)$$

where $Y_g$ is the dataset of the whole group $g$, containing data for each participant in the group, $y_{s \in g}$. Importantly, a separate analysis was performed for each group, to ensure that the participant's posterior probabilities $P(m_s | Y_g)$ were derived from his or her group's distribution. It should be noted that this was possible because the computational top-down modulation family outperformed the other families in all three groups, and no statistical differences were detected between groups with respect to the preferred model architecture (see the "Results" section). Statistical analyses were performed on BMA coupling parameters using one-tailed $t$ tests according to a priori hypotheses, in the three target memory regions (rHIP, cHIP, PC), as well as the wHIP (i.e., four regions in total). Four effects were tested:

1. Control * Group interactions comparing the control effect (predictive - reactive) in PTSD+ with both PTSD− and nonexposed in all four regions (i.e., 8 tests in total):
2. Control effect (predictive - reactive) in all three groups and four regions (i.e., 12 tests in total);
3. Reactive negative coupling in all three groups and four regions (i.e., 12 tests in total);
4. Predictive negative coupling in all three groups and four regions (i.e., 12 tests in total).

To control for Type I error across multiple tests, $p$ values were adjusted for each of these effects, using FDR correction. For completeness, we also computed the Pp of the groups' coupling parameters, as well as the bootstrapped 95% CI of the mean. In addition, we also report Bayes factors (BF) as effect size in Table 1, using a Markov chain Monte Carlo (MCMC) method[69]. BF represent the likelihood of suppression effects for each within-group comparison. Based on this hypothesis, we defined a region of practical equivalence (ROPE) set as a Cohen's d effect size greater than "0.1". The MCMC method generated 90,000 credible parameter combinations that are representative of the posterior distribution. Then, the BF was estimated as the ratio of the proportion of the posterior within the ROPE relative to the proportion of the prior within the ROPE. The conventional interpretation of the magnitude of the BF is that there is substantial evidence for the alternative hypothesis when the BF ranges from 3 to 10, strong evidence between 10 and 30, very strong evidence between 30 and 100, and decisive evidence above 100.

**Imbalance analysis**. We projected neurocomputational markers of predictive and reactive control of intrusive memories onto two orthogonal axes of a polar coordinate system (see Fig. 4b). Angular coordinates were expressed in degrees between −180° and +180° with a 0° reference point at the bottom of the y-axis (i.e., 0° to 180° anticlockwise and 0° to −180° clockwise). The first axis (+45° to −135°) represented predictive control (PC). Negative PC coupling values were projected on the +45° direction, and positive PC coupling parameters onto the opposite −135° direction. The second axis (+135° to −45°) represented reactive control (RC). Negative RC coupling values were projected onto the −45° direction, and positive RC coupling parameters onto the opposite +135° direction.

For each participant, we calculated the resultant force (**RF**) combining predictive and reactive forces. The RF represents the vector sum of a set of forces. Given two forces $F_{PC}$ and $F_{RC}$, characterized by known angles $\alpha_1$ and $\alpha_2$ from 0° on the y-axis of a circle and the $x$ and $y$ Cartesian components (**Fx$_{PC}$**, **Fx$_{RC}$** and **Fy$_{PC}$**, **Fy$_{RC}$**), the RF's Cartesian components can be obtained as follows:

$$\mathbf{F_{Rx}} = Fx_{PC} + Fx_{RC};$$

$$\mathbf{F_{Ry}} = Fy_{PC} + Fy_{RC}; \qquad (17)$$

In our analyses, we focused on the **RF**'s direction, not its magnitude. The **RF** represents the summative effect of predictive and reactive vectors of force. As the two forces were applied in different directions, yet both pointing downward, the 0° position represented the equilibrium point. The more the **RF** approached the 0° direction, the more balance the two forces were. To obtain an imbalance angle (IB)

for each participant, we computed the angle $\theta$ between the **RF** and the 0° position using trigonometry:

$$IB = \theta = \tan^{-1}\left(\frac{F_{Ry}}{F_{Rx}}\right) \qquad (18)$$

Interestingly, both predictive and reactive negative coupling parameters reflected downward, yet orthogonal forces, originating from the same point of application. This illustrates how a unique control system may suppress memory processing according to two independent but complementary processes serving the same function.

A common issue in circular statistics is the arbitrary choice of the 0° position and the sense of rotation, which can lead to misleading conclusions when dealing with multiple measures. The mean angle $\theta$ cannot be computed from the arithmetic mean of all sampled angles. We used the Circular Statistics toolbox in MATLAB[70] to compute the mean angle $\theta$ across participants in each group. Confidence intervals were also computed by bootstrapping the estimation of this group mean 2000 times. Group comparisons were performed using Watson's two-sample tests, a nonparametric version of the two-sample $t$ test for circular data. For all group comparisons, alpha was set at 0.05.

**Reporting summary**. Further information on research design is available in the Nature Research Reporting Summary linked to this article.

## Data availability

All the raw behavioral and imaging data are archived at the GIP Cyceron center in Caen and are part of an ongoing longitudinal research project. Raw behavioral and brain imaging data are available under restricted access due to legal and sponsorship regulation for the current research project. These data can be shared with researchers upon reasonable request, via data request to Dr. Pierre Gagnepain (pierre.gagnepain@inserm.fr). The clinical data and the subject-specific DCM coupling parameters data necessary for the statistical analyses of the current research article have been deposited on the GitHub repository (https://github.com/PierreGagnepain/predictive_control) and are also available on Zenodo (https://doi.org/10.5281/zenodo.6362400)[71].

## Code availability

Computational models were implemented in the TAPAS toolbox (https://www.tnu.ethz.ch/de/software/tapas). Preprocessing of fMRI data and first-level DCM analysis were performed with SPM12 (https://www.fil.ion.ucl.ac.uk/spm/; version DCM12.5 revision 7479). The log-family evidence was computed using the MACS toolbox (https://github.com/JoramSoch/MACS/releases/tag/v1.3), and Bayesian model comparisons were performed with the VBA toolbox (https://mbb-team.github.io/VBA-toolbox/). Codes for implementing model falsification, parameter and model recovery, as well as computational DCM to study predictive control, is available on GitHub (https://github.com/PierreGagnepain/predictive_control).

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

## Acknowledgements

We thank all the people who volunteered to take part in this study and the victim associations that supported this project. We thank the medical doctors (especially M. Mialon and E. Duprey) and the staff at the Cyceron biomedical imaging platform in Caen. We also thank the researchers; psychologists M. Deschamps, P. Billard, B. Marteau, R. Coppalle, and C. Becquet; technicians; and administrative staff at U1077 (Caen), at "Program 13-Novembre" in Paris, at INSERM "Délégation Régionale Nord-Ouest" (Lille) and "Pôle Recherche Clinique"(Paris). We thank Jean-François Démonet for comments and feedbacks on this manuscript. We thank Elizabeth Portier for final English editing of the manuscript. This study was funded by the French Commissariat-General for Investment (CGI) via the National Research Agency (ANR) and the "Program d'investissement pour l'Avenir (PIA)." The study was realized within the framework of "Program 13-Novembre" (EQUIPEX Matrice) headed by D.P. and F.E. This program is sponsored by the CNRS and INSERM and supported administratively by HESAM Université, bringing together 35 partners (see www.memoire13novembre.fr). G.L. is funded by a Ph.D. fellowship from the Normandy Region and Normandy University.

## Author contributions

J.D., D.P., F.E., and P.G. designed the study. P.G. and G.L. conceptualized and implemented the computational model. P.G., J.D., D.P., F.E. obtained the financial support. C.P., A.M., and T.V. performed the data acquisition and F.F. managed and coordinated the research activity planning and execution. F.V. and V.d.L.S. supervised the MRI data collection and medical interviews. V.d.L.S. supervised the medical aspects of the study, and J.D. supervised the SCID interviews and psychiatric examinations. G.L. and P.G. analyzed the behavioral and functional data. G.L. and P.G. wrote the original draft. P.G. supervised the research. All the authors reviewed and edited the manuscript.

## Competing interests

The authors declare no competing interests.
