## [Peer Review File · Nature Communications]

Altered predictive control during memory suppression in PTSDEditorial Note: This manuscript has been previously reviewed at another journal that is not operating a transparent peer review scheme. This document only contains reviewer comments and rebuttal letters for versions considered at *Nature Communications*.

REVIEWERS' COMMENTS

Reviewer #1 (Remarks to the Author):

This is a transferred manuscript twice previously reviewed by me. In this third version, the authors have satisfactorily addressed my remaining two small concerns.

Reviewer #3 (Remarks to the Author):

The present study is part of a more general and very important work from Dr Gagnepain's team dedicated to the neurobiological bases of traumatic memory formation and persistence. More specifically, this team produced significant scientific contributions indicating that PTSD and its persistence could be due to a deficit of memory inhibition and that a dysfunctional hippocampus could be at the origin of this "forgetting disorder".

In the present study, the authors combined computational modeling and brain connectivity analyses to reveal how individuals exposed and non-exposed to the 2015 Paris terrorist attacks formed and controlled beliefs about future intrusive re-experiencing during a memory suppression task. They show that PTSD patients formed aberrant beliefs and used them excessively to control hippocampal activity. Unlike resilient and unexposed subjects in whom an optimal balance between reactive memory suppression mechanisms and predictive control is observed, PTSD patients present an imbalance of these processes with an exacerbated/aberrant predictive control and a reduced reactive control. An increased negative influence of the dlPFC (control system) on memory systems, which is observed in resilient subjects and subjects not exposed to trauma during attempts to suppress intrusive memories, is not found in PTSD patients. In addition, this imbalance is linked to avoidance, but not to general disturbances such as anxiety or negative affect.

The quality of the present study, both conceptually and methodologically, the reliability of the results and high quality of the multiple analyses presented, as well as the relevance of the scientific issue addressed (the discovery of a new pathological brain mechanism of PTSD) make this research work a remarkable scientific advance in the field of traumatic memory. Furthermore, the authors have very seriously and very convincingly responded point by point to the questions and concerns of the three previous reviewers (cf. new analyzes and significant reworking of the manuscript text).

Accordingly, I strongly recommend its publication in Nature Communications.

REVIEWERS' COMMENTS

Reviewer #1 (Remarks to the Author):

This is a transferred manuscript twice previously reviewed by me. In this third version, the authors have satisfactorily addressed my remaining two small concerns.

We would like to express our deepest gratitude to the reviewer for the time taken to evaluate and assess our latest response. We know that reviewing is a great deal of effort, and we are quite grateful for the reviewer scrupulous consideration of our manuscript across the different stages of the revisions process. We are particularly grateful for the useful feedbacks and suggestions that helped us improving the clarity of the presentation of our methods and results.

Reviewer #3 (Remarks to the Author):

The present study is part of a more general and very important work from Dr Gagnepain's team dedicated to the neurobiological bases of traumatic memory formation and persistence. More specifically, this team produced significant scientific contributions indicating that PTSD and its persistence could be due to a deficit of memory inhibition and that a dysfunctional hippocampus could be at the origin of this "forgetting disorder". In the present study, the authors combined computational modeling and brain connectivity analyses to reveal how individuals exposed and non-exposed to the 2015 Paris terrorist attacks formed and controlled beliefs about future intrusive re-experiencing during a memory suppression task. They show that PTSD patients formed aberrant beliefs and used them excessively to control hippocampal activity. Unlike resilient and unexposed subjects in whom an optimal balance between reactive memory suppression mechanisms and predictive control is observed, PTSD patients present an imbalance of these processes with an exacerbated/aberrant predictive control and a reduced reactive control. An increased negative influence of the dlPFC (control system) on memory systems, which is observed in resilient subjects and subjects not exposed to trauma during attempts to suppress intrusive memories, is not found in PTSD patients. In addition, this imbalance is linked to avoidance, but not to general disturbances such as anxiety or negative affect.

The quality of the present study, both conceptually and methodologically, the reliability of the results and high quality of the multiple analyses presented, as well as the relevance of the scientific issue addressed (the discovery of a new pathological brain mechanism of PTSD) make this research work a remarkable scientific advance in the field of traumatic memory. Furthermore, the authors have very seriously and very convincingly responded point by point to the questions and concerns of the three previous reviewers (cf. new analyzes and significant reworking of the manuscript text). Accordingly, I strongly recommend its publication in Nature Communications.

We are very grateful to the reviewer for these positive comments on this work. We thank the reviewer for the time taken to examine both our manuscript and the previous exchanges with other reviewers.